# Adaptive hierarchical origami-based metastructures

Yanbin Li [1,4] ✉, Antonio Di Lallo[1,4], Junxi Zhu[1], Yinding Chi[1], Hao Su [1,2,3] ✉ & Jie Yin [1] ✉

Shape-morphing capabilities are crucial for enabling multifunctionality in both biological and artificial systems. Various strategies for shape morphing have been proposed for applications in metamaterials and robotics. However, few of these approaches have achieved the ability to seamlessly transform into a multitude of volumetric shapes post-fabrication using a relatively simple actuation and control mechanism. Taking inspiration from thick origami and hierarchies in nature, we present a hierarchical construction method based on polyhedrons to create an extensive library of compact origami metastructures. We show that a single hierarchical origami structure can autonomously adapt to over $10^3$ versatile architectural configurations, achieved with the utilization of fewer than 3 actuation degrees of freedom and employing simple transition kinematics. We uncover the fundamental principles governing theses shape transformation through theoretical models. Furthermore, we also demonstrate the wide-ranging potential applications of these transformable hierarchical structures. These include their uses as untethered and autonomous robotic transformers capable of various gait-shifting and multidirectional locomotion, as well as rapidly self-deployable and self-reconfigurable architecture, exemplifying its scalability up to the meter scale. Lastly, we introduce the concept of multitask reconfigurable and deployable space robots and habitats, showcasing the adaptability and versatility of these metastructures.

Versatile shape-morphing capability is crucial for enabling multi-functionality in both biological and artificial systems, allowing them to adapt to diverse environments and applications[1–3]. For example, the mimic octopus can rapidly transform into up to 13 distinct volumetric shapes, mimicking various marine species[1]. In the realm of artificial systems, there has been a range of strategies proposed to create shape-morphing structures, including continuous forms of beams, plates, and shells[4–6], bar-linkage networks or mechanical kinematic mechanisms[7–13], folding or cutting-based origami/kirigami structures[12,14–22], and reconfigurable robotic structures composed of assembled magnetic or jointed modules[23–28]. These structures have found broad applications in transformable architecture[21,29], reconfigurable robotics[25,30], biomedical

devices[8,31], flexible spacecraft[32,33], multifunctional architected materials[20,34], reprogrammable shape-morphing matter[6,35,36], as well as deployable structures that can undergo dramatic volume change for convenient storage and transport[15,29,32,33,37–39].

However, despite these advancements, artificial shape-morphing structures have yet to rival their biological counterparts in terms of the diversity of attainable volumetric shapes, as well as the efficiency and autonomy with which such versatile shape morphing can be achieved through simple actuation and control[6,23,24,26,27,35,36]. One of the primary challenges resides in the tradeoff between theoretically allowable versatility of shape-morphing, which encompasses the quantity and diversity/type of reconfigured shapes, and practical controllability in

[1]Department of Mechanical and Aerospace Engineering, North Carolina State University, Raleigh, NC 27606, USA. [2]Lab of Biomechatronics and Intelligent Robotics, Joint NCSU/UNC Department of Biomedical Engineering, Raleigh, NC, USA. [3]University of North Carolina at Chapel Hill, Chapel Hill, NC, USA. [4]These authors contributed equally: Yanbin Li, Antonio Di Lallo. ✉e-mail: yli255@ncsu.edu; hsu4@ncsu.edu; jyin8@ncsu.edu

terms of actuation. For instance, while previously reported structures[11,23,24,26–28,36] have demonstrated the ability to change into a vast number of distinct shapes, they often require exceedingly complex actuation and control systems. This complexity can render the shape morphing process tedious, time-consuming, and energy-inefficient. On the other hand, certain structures may exhibit simpler reconfiguration kinematics[3,5–7,15,31,40–42], enabling them to feasibly attain desired shapes. However, their specified structural forms may largely limit the achievable reconfigured shapes within few specific categories. These challenges, along with others such as complex reconfiguration kinematics, poor re-programmability, lack of inverse design capability, and limited functionality of the reconfigured shapes, as summarized in Supplementary Table 1, could considerably impede the broad applications of shape-morphing structures in areas such as reconfigurable architecture, metamaterials, and robotics (see more details in Supplementary Note 1). The versatility of shape morphing is intricately linked to a structure's mobility, i.e., the number of degrees of freedom (DOF). Theoretically, structures with a higher number of DOFs tend to exhibit greater versatility in shape morphing[11,23–27,36]. However, this very versatility in theory often makes it exceedingly difficult to actuate structures with higher DOFs, considering the potential need for distributed actuation of each DOF[25].

Conventional rigid mechanism-based origami structures, constrained by their folding interconnections, are limited to morphing between their original and compact states due to one single DOF. This limitation simplifies actuation and deployment but sacrifices the potential for achieving a variety of shapes[12,13,16,18,22,38,40,43]. To address this limitation, recent advances have introduced modular origami metastructures composed of assembled polyhedron-shaped modules[26,36,39], such as cubes and tetrahedrons, etc. These structures offer more than four mobilities. For example, recent studies demonstrated that a single unit cell consisting of six extruded cubes could transform into four different configurations using four distributed pneumatic actuators to control folding angles[39]. However, when scaling up to a $4 \times 4 \times 4$ periodic meta-structures to achieve similar transformations, it requires a staggering 96 distributed actuators for each DOF[39], resulting in low actuation efficiency. More recently, we proposed shape-morphing planar kinematic origami/kirigami modules composed of a closed-loop connection of eight cubes[36]. These modules can be manually transformed into over five different configurations via kinematic bifurcation. When assembled into a $5 \times 5$ array, they theoretically offer over 10,000 mobilities through bifurcation[36]. However, practically, they pose grand challenges in terms of actuation and control. Similarly, discrete kinematic cube-based modules are often assembled into lattice, chain, or hybrid architectures and used in robotic structures with higher DOFs for multifunctional modular reconfigurable robots[25]. Although these modular origami and robotic structures offer enhanced shape-morphing capabilities, they typically require control and actuation systems for each module. This complexity results in lengthy and intricate reconfiguration steps, as well as complex and time-consuming actuation, morphing kinematics, and reconfiguration paths, primarily due to their redundant DOFs[11,25–27,35,36] (Supplementary Table 1 and related discussions in Supplementary Note 1).

Drawing inspiration from planar thick-panel origami[12,18,22,36] and hierarchical materials/structures[44–47] in nature and engineering, here, we propose leveraging hierarchical architecture of spatial closed-loop mechanisms interconnected both within (locally) and across (globally) each hierarchical level to address the versatility-actuation tradeoff in an example system of highly reconfigurable hierarchical origami metastructures. As illustrated in Fig. 1a, a base or level-1 structure is a spatial closed-loop mechanism consisting of $n$ rigid linkages and $n$ rotational hinges, an $n$R looped mechanism. Simply replacing each rigid linkage in a $k$R looped mechanism with the level-1 structure creates a level-2 "$k$R" spatial looped flexible mechanism (Fig. 1b), since

each linkage becomes an $n$R looped mechanism, with $k$ being the number of rotational hinges at level 2 (note that $k$ is not necessarily equal to $n$). The rotary hinges can employ origami line folds and the rigid links can take variously shaped structural elements, such as thick plates and polyhedrons (e.g., cubes, triangular or hexagonal prisms) (Fig. 1c). The polyhedrons can be combinatorically connected at their edges using rotary hinges at each hierarchical level, offering extensive design space for diverse reconfigurable hierarchical metastructures (Fig. 1d–f and Supplementary Note 2).

We demonstrate the unprecedented properties of the metastructures arising from their hierarchical architecture of spatial closed-loop mechanisms. We find that hierarchical closed-loop mechanisms naturally introduce intricate geometric constraints that dramatically reduce the number of active DOFs required for shape morphing, even when involving a large number of structural elements (Fig. 1f, g). Benefiting from this hierarchical coupling of closed-loop mechanisms, we show that these hierarchical origami metastructures can be efficiently actuated and controlled while achieving a wealth of versatile morphed shapes (over $10^3$) through simple reconfiguration kinematics with low actuation DOF ($\leq 3$) (Fig. 1g). The proposed construction strategy unlocks a vast design space by orchestrating combinatorial folding both within and across each hierarchical level, relying on spatial closed-loop bar-linkage mechanisms. It effectively overcomes the intrinsic limitations in our previous ad-hoc shape-morphing designs with similar structural elements[36], including geometric frustrations, large number of DOF, and a lack of generalizability due to the use of units with specific shapes[13] (see Supplementary Note 1.2 for detailed comparison). Compared to the state-of-the-art shape-morphing systems[7–11,14,16,18,22,24–30,32,36,37,39,40,43,48], our combinatorial and hierarchical origami-inspired design shows superior multi-capabilities, including high reconfiguration and actuation efficiency (requiring less time and fewer transition steps and actuations), simple kinematics and control, high (re)-programmability, a large number of achievable shapes, and potential multi-functionalities (see Supplementary Note 1.1 and Supplementary Table 1 for detailed comparison). We explore the underlying science of versatile shape morphing and actuation in the hierarchical origami metastructures, as well as their applications in self-reconfigurable robotics, rapidly self-deployable and transformable buildings, and multi-task reconfigurable space robots and infrastructure.

## Results
### Hierarchical origami-based shape-morphing structures with combinatorial design capability

Figure 2a–c and Supplementary Figs. 1–3 illustrate the hierarchical approach employed to construct a category of planar thick-panel origami-based shape-morphing structures. In Fig. 2a, the level-1 structure represents an over-constrained rigid spatial bar-linkage looped mechanism, characterized by the number of linkages being equal to or greater than the connected bars. This structure consists of $n$ (where $n = 4, 6, 8$) rigid cubes (Fig. 2a, i) serving as linkages interconnected by $n$ hinge joints (i.e., line folds) at cube edges functioning as rotatable bars (see details in Fig. 2a, ii)[36]. These hinges are highlighted by yellow lines in Fig. 2a, iii. An example of a level-1 structure with $n = 8$ is shown in Fig. 2a, ii and iii, while additional examples with $n = 4$ and 6 are depicted in Supplementary Fig. 1a–c.

The connectivity between the cubes, namely the placement of the joints, dictates the spatial folding patterns of the structure (Supplementary Figs. 1 and 2). Broadly, the deployment follows four fundamental structural motifs, defined here as the mechanism-based connecting systems used to construct each leveled structure: one 2R chain-like mechanism and three 4R, 6R[10,18,22], or 8R closed-loop mechanisms[41], where $n$R denotes mechanisms with $n$ rotational links and $n$ rotatable (R) joints (see Supplementary Fig. 3, and detailed

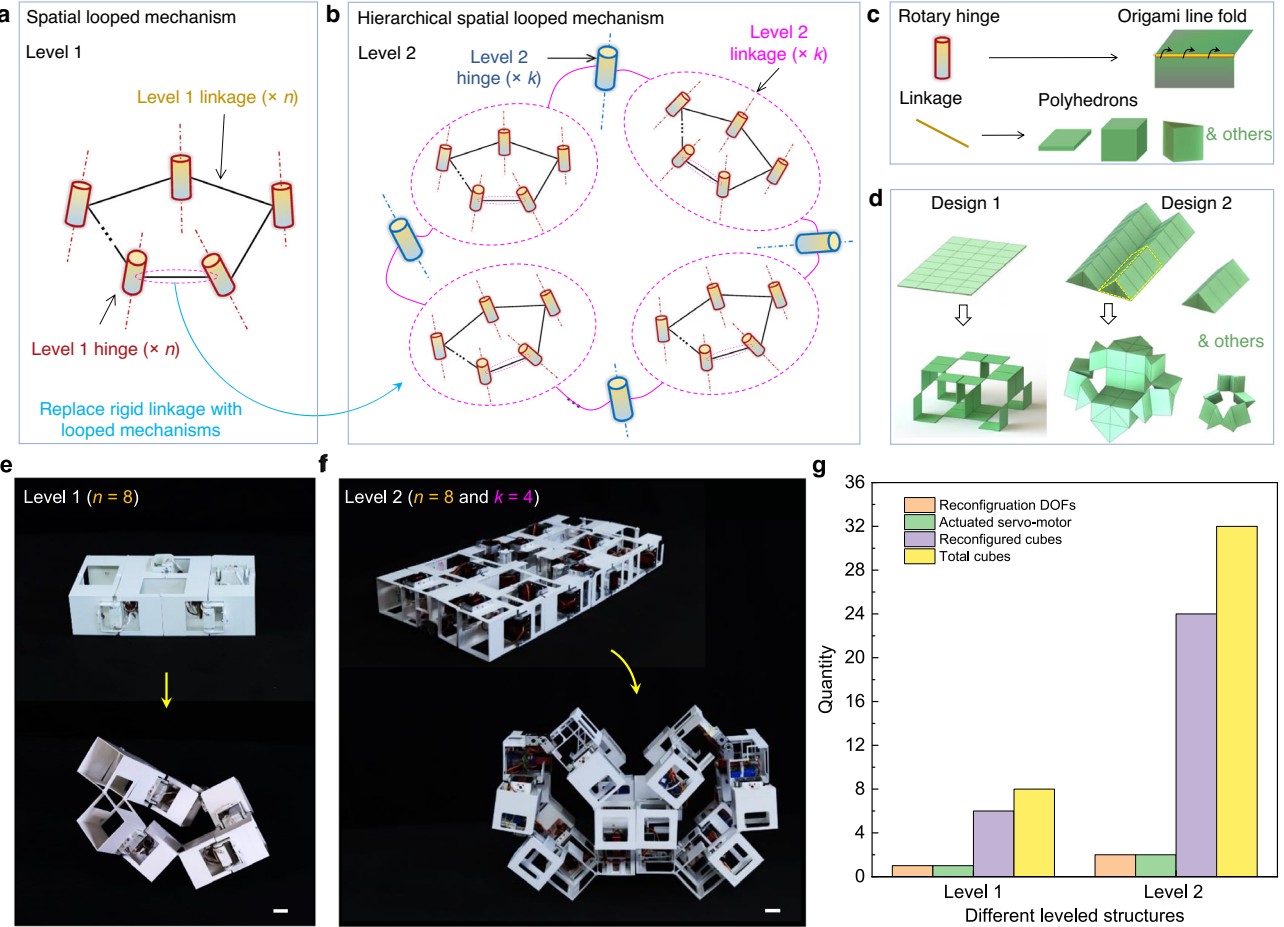

**Fig. 1 | Overview of the construction and advantages of hierarchical origami-based shape-morphing metastructures.** Schematic illustrations of a level-1 metastructure composed of an $n$R spatial looped mechanism with $n$ rotary hinges and $n$ rigid linkages (**a**) and a level-2 metastructure composed of a "$k$R" spatial looped mechanism at level 2 and $n$R looped mechanisms at level 1 (**b**). **c** The designs of rotary hinges and rigid linkages in the forms of respective origami line fold and different polyhedrons. **d** Illustration of two types of reconfigurable metastructures using planar and spatial tessellation of thin plates and prims, respectively. Examples of 3D-printed prototypes of self-reconfigurable level-1 (**e**) and level-2 (**f**) origami-based robotic metastructures actuated by electrical servomotors. Scale bar: 3 cm. The level-1 and level-2 metastructures are composed of closed-loop connections of 8 and 32 cubes, respectively. **g** Demonstration of the advantages of hierarchical looped mechanism in creating self-reconfigurable metastructures with versatile shape morphing under fewer reconfiguration DOFs (actuated servomotors) than 3.

definitions in Supplementary Note 3). For two adjacent cube faces, four potential edge locations exist to accommodate hinge joints (Fig. 2d, i). Consequently, a structure with $n$ cubes theoretically allows for $4^n$ combinatorial sets of connections, offering an extensive design space for level-1 structures (Supplementary Fig. 4). Specially, we define this multiple design possibility by the placement of hinge joints in all leveled structures as their combinatorial design capability. As illustrated later, the combinatorial design capability of our proposed systems can be considerably expanded given the structural asymmetries and the multiple choices of structural motifs. Depending on the chosen connectivity, level-1 structures composed of $n$ cubes exhibit an initial maximum number of 2 ($n = 4$), 3 ($n = 6$) and 5 ($n = 8$) DOFs (Fig. 2e), which can be utilized for morphing into a diverse array of distinct 3D architected structures (as exemplified in Fig. 2a, iv, and further elaborated in Supplementary Fig. 1 and Supplementary Movie 1).

By substituting the higher-level linkages with the lower-level basic or hierarchical structures (e.g., Fig. 2a–c, i–iii and Supplementary Fig. 3a, b) in the four fundamental structural motifs (2R, 4R, 6R, and 8R), we can create a class of flexible spatial hierarchical mechanism-based origami structures by combinatorically choosing any type of the $n$R linkages as different-level structural motifs (Supplementary Fig. 3c).

Notably, the term "flexible spatial mechanism" refers to mechanical mechanisms with bars and linkages arranged in 3D space, where the length of linkages is not fixed and varies during reconfiguration. For example, Fig. 2c, ii illustrates a level-3 structure comprising 8R linkages at level 1, 4R linkages at level 2, and 2R linkages at level 3, denoted as <8R, 4R, 2R>. The sequence from left to right corresponds to the structural motifs used from lower-level structure to higher-level structure.

The associated level-2 structure is depicted in Fig. 2b and denoted as <8R, 4R>. Additional examples of hierarchical origami structures with varying numbers of cubes at level 1 are presented in Supplementary Figs. 2, 5 and 6. Upon deployment, these structures can continuously transform into a multitude of intricate architected forms featuring internal structural loops (ISLs): internal voids within reconfigured architected structures enclosed by boundary structural components (as illustrated in Fig. 2a–c, iv, Supplementary Figs. 5c and 6). These ISLs efficiently facilitate different-level kinematic bifurcations, where a singular configuration state triggers a sudden increase in structural DOFs, leading to additional subsequent reconfiguration branches. This is in sharp contrast to the counterparts composed of four cubes at level 1, which are primarily limited to simple chain-like configurations (Supplementary Fig. 2a) despite having a greater

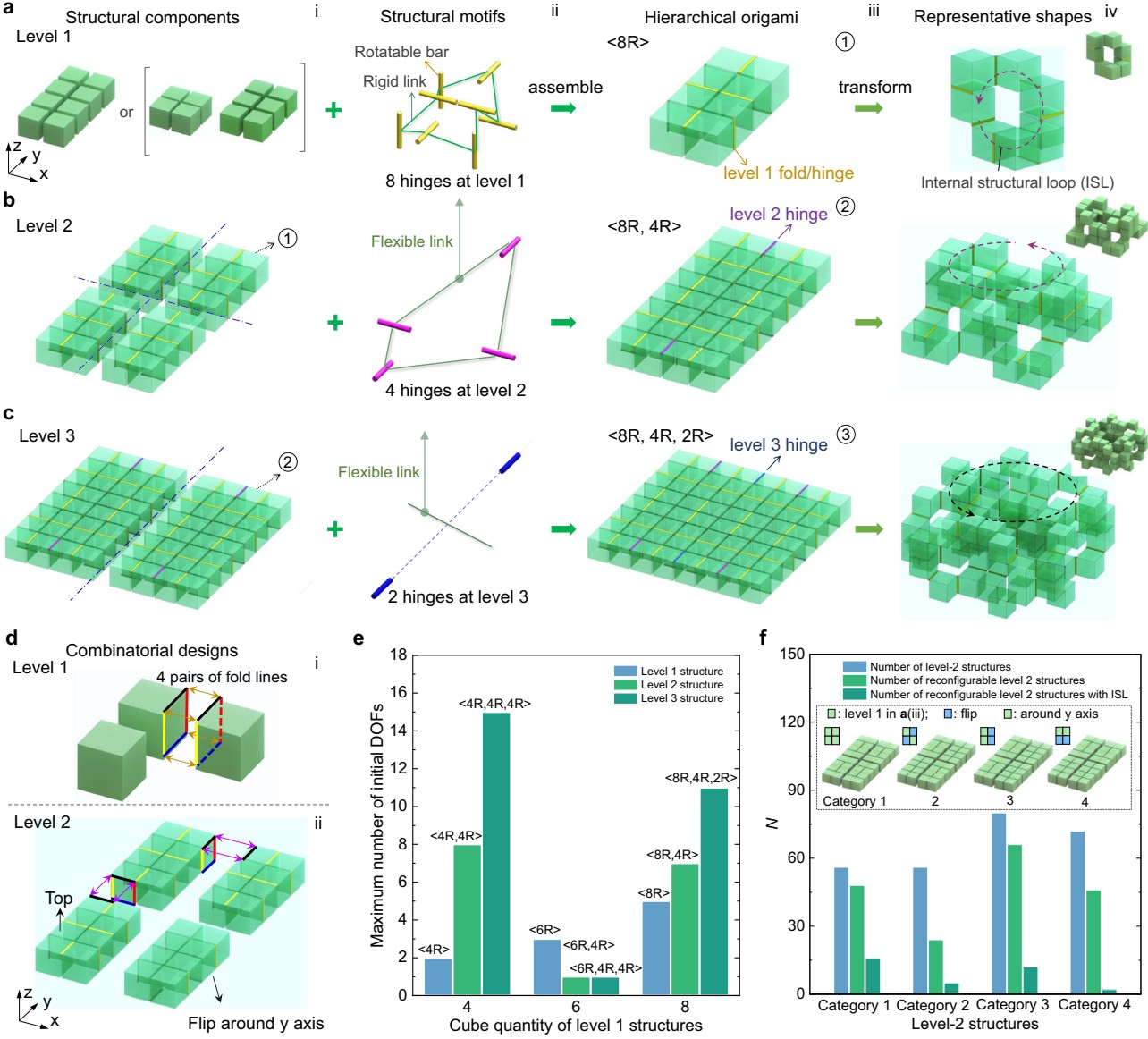

**Fig. 2 | Design of hierarchical origami-based shape-morphing metastructures.**
a–c Schematics of constructing level-1 (**a**), level-2 (**b**), and level-3 (**c**) reconfigurable and deployable structures using hierarchical closed-loop rigid bar (line hinges)-linkage (cubes) mechanisms (column ii) as different-leveled structural motifs (column iii). The representative morphed architectures with internal structural loops (ISLs) are shown in column iv. **d** Schematics of selected combinatorial designs by either combinatorially hinging two adjacent cubes at one of the four cube edge pairs at level 1 (i) and level 2 (ii) or flipping any level-1 structure with asymmetric hinge locations on top and bottom surfaces (ii) or combined. **e** Comparison of the maximum initial structural DOFs of different hierarchical structures composed of 4, 6, and 8 cubes at level 1. **f** Comparison of the combinatorially designed four categories of level-2 structures in (**b**) (insets and Supplementary Fig. 6) on the number of combinatorial level-2 hinge connections, reconfiguration modes, and morphed configurations with ISLs.

number of initial DOFs in the hierarchical structures of <4R, 4R> and <4R, 4R, 4R> (Fig. 2e).

Moreover, the design space of hierarchical structures can be considerably expanded by combinatorially (1) adjusting the connectivity at higher-level bars (Fig. 2d, ii) and (2) manipulating structural asymmetries at the lower-level linkages given the asymmetric patterned joints on the top and bottom surfaces across the thickness, e.g., simple upside-down flipping (see Fig. 2d, ii for an example of level 2 structure). As an illustration, the insets in Fig. 2f and Supplementary Fig. 5 show four selected categories of combinatorial <8R, 4R> level-2 structures created by flipping the level-1 8R linkages and modifying the connections at the level-2 joints. By employing combinatorial design strategies involving mechanism hierarchy, spatial fold patterning across multiple levels of bars, and folding asymmetries in the linkages, we can generate an extraordinary vast design space encompassing

millions of configurations, even within a simple level-2 structure (see analysis in Supplementary Note 2).

Compared to state-of-the-art 2D[14,16,40,48] and 3D origami designs[12,18,22,36,42] including our previous ad-hoc design of specific tessellated closed-loop mechanism of cubes[36], this hierarchical approach offers several advantages: Firstly, it largely broadens the range of designs by allowing combinatorial connections within and across each hierarchical mechanism, which are either disabled or severely limited in previous studies[12,18,22,36,42]. Secondly, it effectively avoids geometric frustration in our previous ad-hoc designs[36], which refers to structural constraints arising from deformation incompatibility during deployment[44,45]. This avoidance is made possible by the compatible reconfigurations of differently leveled spatially looped mechanisms (Fig. 2a–c, ii). Thirdly, this fundamental design principle establishes a versatile structural platform that can be applied to various shaped

building blocks, overcoming the limitations in our previous ad-hoc designs[36] and other studies associated with specific structural elements[22,26,36,38,39,42]. Fourthly, it possesses the intrinsic benefit of structural hierarchy[46,47,49], favoring higher-level structures with greater diversity and quantity of actuated reconfigured shapes under simple control and actuation.

Within this extensive design space, designs of particular interest are those that exhibit high reconfiguration capabilities via collision-free kinematic paths involving only a few active structural DOF during shape-changing processes. Such designs enable rich shape-morphing capability with simple and reliable control. After comparison (Supplementary Note 2), we identified an optimal category composed of four identical <8R> type of level-1 structures (see Category 1 in Fig. 2f and Supplementary Fig. 5b, with detailed definitions provided in Supplementary Notes 2 and 3) to showcase their extensive shape-morphing behavior under few active DOF. These designs boast the highest structural symmetries and the largest number of ISLs, facilitating bifurcation and shape diversity (Fig. 2f).

## Continuously evolving versatile shape morphing

Figure 3a provides a comprehensive view of the shape-morphing configurations diagram of one exemplary optimal <8R, 4R> level-2 structure selected from Category 1 in Fig. 2b (see Fig. 3a, i for its hierarchical design details). These structures were fabricated by assembling the 3D-printed rigid square facets (in white) into hollow cubes via interlocking mechanisms and flexible printed line hinges made of rubber-like materials (in black) (Supplementary Fig. 7a, see "Methods" and Supplementary Movie 2 for details). This design not only facilitates straightforward assembly but also allows for easy disassembly and reassembly of facets into hierarchical structure (Supplementary Fig. 7b–d). For clarity, configurations with folding angles that are multiples of 90° are displayed since these angles correspond to kinematic bifurcations, as discussed later.

With the inherent capacity for versatile shape changes provided by the level-1 linkage structure (Supplementary Fig. 7b), the level-2 structure can continuously evolve, adopting various representative complex architectures along multiple reconfiguration paths (indicated by different colored lines in Fig. 3a). Notably, these shapes bear a striking resemblance to trucks, trophies, tunnels, shelters, and various architectural structures (see more details in Supplementary Fig. 8 and representative reconfiguration processes in Supplementary Movie 3).

To systematically represent all reconfigured shapes and their corresponding shape transitions in Fig. 3a (ii), we employ a data-tree-like diagram (Supplementary Fig. 9), inspired by graph theory used in computer science to elucidate logical relationships among adjacent data nodes[50] (Supplementary Note 4). In this diagram, both nodes and line branches are assigned specific physical meanings, signifying individual reconfigured shapes and the relative shape-morphing kinematic pathways connecting them. As shown in Fig. 3a and Supplementary Fig. 9, starting from a compact state (node $M_A$), the analyzed level-2 structure can traverse a closed-loop shape-morphing path (termed reconfiguration loop 1, RL-1, or a parent loop). Along this path, it transitions from simple chain-like structures (e.g., node $M_A \rightarrow M_B \rightarrow M_C \rightarrow M_D$) to intricate architectures featuring ISLs (e.g., node $M_D \rightarrow M_E \rightarrow M_F$). Subsequently, starting from node $M_E$ with ISLs, it can further transform into nodes $M_F$, $M_5$, $M_6$, or return to $M_D$). Theoretically, this continuous evolution in shape arise from the varying link lengths of the flexible level-2 linkage as line folds exhibit changing folding angles (Fig. 3b and Supplementary Fig. 11, see the analysis in Supplementary Notes 5–7.1, which examines length variations in level-2 links during two representative shape-morphing processes from node $M_D$ to $M_E$ and from node $M_E$ to $M_F$).

Benefitting from both chain-like and closed-loop mechanisms embedded in the morphed structural configurations, the parent loop gives rise to several subtrees (e.g., at node $M_A$, $M_B$ or $M_2$, $M_E$, and $M_F$). These subtrees, in turn, branch into more paths through kinematic bifurcations (e.g., at node $M_{11}$, $M_{15}$, and $M_{17}$), as depicted in the inset of Fig. 3b. These bifurcations can be accurately predicted based on the number of null eigenvalues $v_{kk}$ in the kinematics model (see Supplementary Note 7.2 for detailed theoretical analysis). Importantly, node $M_6$ and node $M_{10}$, located in different subtrees, are interconnected to form another reconfiguration loop (i.e., RL-2). This allows for direct transformation between two configurations or nodes that traverse different subtrees efficiently, without the need to return to the initial configuration and repeat redundant transforming steps, as required in previous reconfigurable structures[11,14,16,23–26,28,36]. Comparable hierarchical transition tree structures featuring bifurcated branches and interconnected nodes are observed in most of the four categories of other combinatorial <8R, 4R> level-2 structures (Supplementary Fig. 11). These structures are obtained by rearranging multilevel joint locations on top or bottom surfaces or by flipping the level-1 linkage (as seen in the level-2 representative in Fig. 3d and Supplementary Fig. 7d, e). Consequently, a multitude of versatile and distinct morphed configurations are generated (Supplementary Figs. 12 and 13) based on differing hinge connectivity.

Additionally, for all combinatorial designs (Supplementary Fig. 5b), we observed that the number of reconfiguration paths increases approximately linearly with the number of bifurcated nodes or configurations (Fig. 3c). Notably, starting from a defined fold pattern, the same level-2 structure can generate nearly $10^3$ reconfiguration paths with approximately 100 bifurcation nodes, thereby bestowing extensive shape-morphing capabilities (see analysis in Supplementary Note 8). In comparison to previous designs[12,16,18,22,26,29,31,38–40,43,48] that offer only a few shape-morphing paths from a defined fold pattern, our hierarchical design strategy enables a high number ($N \sim 10–10^3$) of kinematic transitions, demonstrating substantial versatility in generating numerous shapes and architectures.

Given that each reconfigured shape in Fig. 3a is defined by internal fold rotation angles that are multiples of 90°, we can accurately represent each shape by collecting spatial vectors $v$ of the body center coordinates of all structural elements into a shape matrix $\mathbf{M}$ (see "Methods" for details). This matrix takes the explicit form $\mathbf{M} = (v_1, v_2, v_3, ..., v_n)$ (with $n = 32$ for the level-2 structures shown in Fig. 3 and Supplementary Fig. 5, see "Methods" and Supplementary Note 4 for details). Consequently, we can systematically annotate all reconfigured shapes in Fig. 3a using their corresponding shape matrices $\mathbf{M}_k$ (with k as the shape index, see inset in Fig. 3a and Supplementary Fig. 9). Once the initial shape matrix $\mathbf{M}_A$ is known, we can theoretically determine all the reconfigured shapes of the level-2 structure in Fig. 3a accordingly (see "Methods" for details). Importantly, this annotation approach is generalizable and can be applied to all other hierarchical origami metastructures presented in this work.

Remarkably, despite the level-2 structure's total of 36 joints, only a small number of them are needed to drive the shape-morphing process, referred to as active reconfiguration DOF (Fig. 3e). For example, when considering the multistep shape-morphing process from node $M_D$ to node $M_A$, i.e., $M_D \rightarrow M_E \rightarrow M_F \rightarrow M_A$ in Fig. 3a, it exhibits only 2, 2 and 1 DOF, respectively, even though it involves the rotation of 16, 8, and 24 joints (Fig. 3e and more details in Supplementary Figs. 14 and 15). This is in contrast to our previous ad-hoc design of cube-based reconfigurable metastructures[36]. Despite the presence of multiple closed-chain loops, they often function as independent units that barely couple with each other during shape morphing due to the specific architecture design of these metastructures, which results in high mobilities over 10,000[36], making it impossible for control and actuation. In contrast, the reduction in active joints in this work is due to the specific

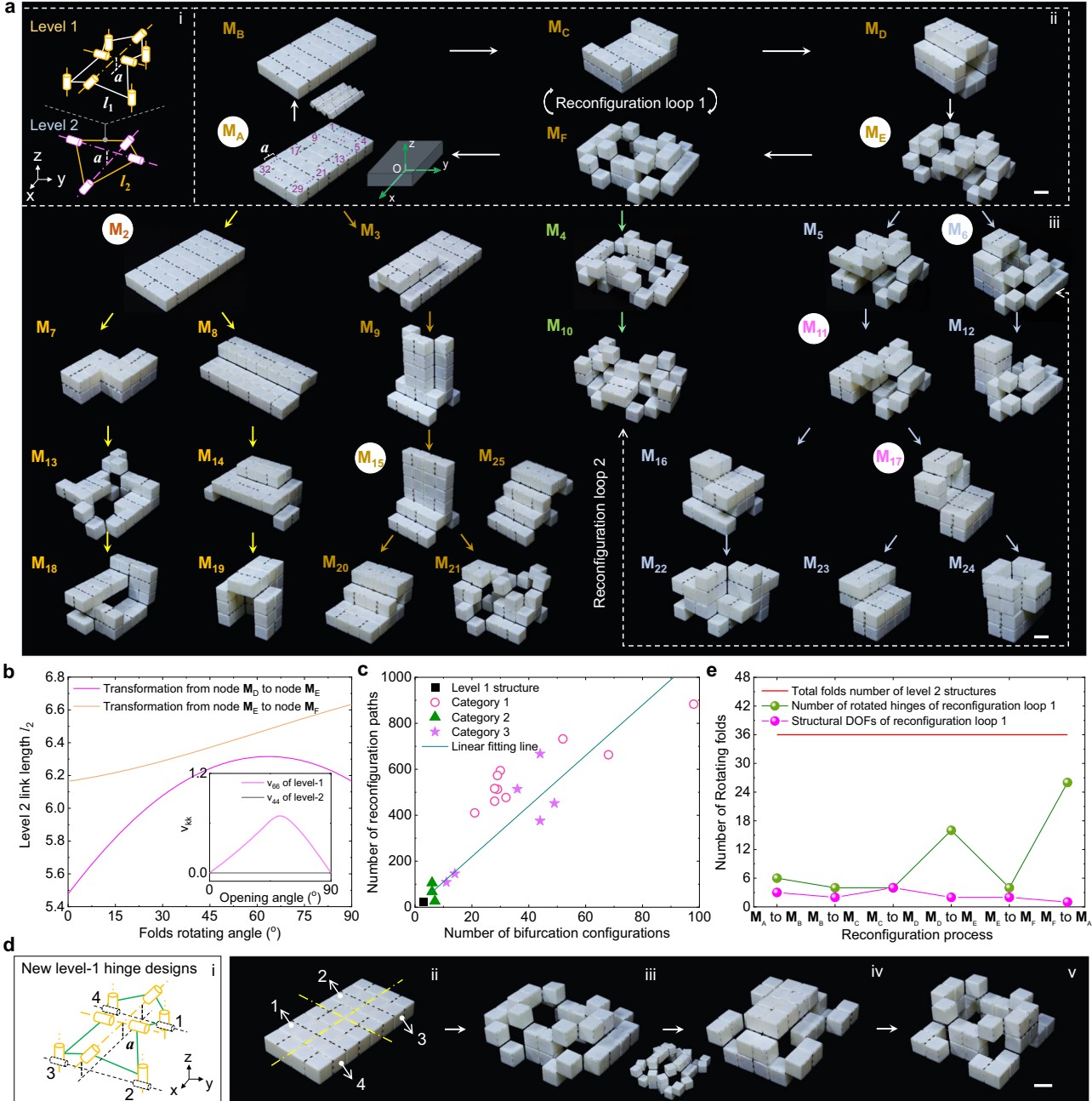

**Fig. 3 | Continuous shape morphing of an optimal level-2 structure. a** Shape-morphing configurations diagram in the 3D-printed prototype exhibiting hierarchical transition tree-like features. The branches in the transition tree of represent the bifurcated configurations. Scale bar: 3 cm. **b** The variation of flexible level-2 link length with the opening angle of hinges during the shape transition from node $M_D$ to $M_E$, and node $M_E$ to $M_F$ in reconfiguration loop 1 in (**a**). Inset shows the eigenvalues $v_{kk}$ as a function of the rotating angle in both level-1 and level-2 structures. **c** The relationship between the number of reconfiguration paths and the number of kinematic bifurcation configuration states for the combinatorially designed category I–III level-2 systems. **d** One selected combinatorial design of the shape-morphing level-2 structures by rearranging the level-1 hinges (i), and some of its representative reconfigured shapes (ii–v). Scale bar: 3 cm. **e** Comparison among the total number of hinges, the number of rotated joints, and the number of reconfiguration DOFs during the reconfiguration loop from node $M_A$ to $M_F$ and back to $M_A$ in (**a**).

interconnectivity of the looped level-1 and level-2 structures as geometric constraints, which dramatically reduces the number of active joints required while enabling high reconfigurability. Additionally, the multilevel closed-loop interconnectivity simplifies the control of shape-morphing paths in terms of simple transition kinematics, as demonstrated below.

## Simple transition kinematics during shape morphing

The transition kinematics describes the quantitative relationship among the folding angles during the shape morphing of

hierarchical structures. In Fig. 4a, we utilize the transformation matrix $\mathbf{T}(d, \gamma)$ to describe the relative spatial relationship of the four links, where d is the shortest distance between adjacent joints, and $\gamma$ is the opening angle between adjacent cube-based links, as shown in Fig. 4b, c and Supplementary Fig. 16. For a looped mechanism, it holds that $\sum_{i=1}^{m} \mathbf{T}_i = \mathbf{I}$, where $m = 8$ and $m = 4$ for the level-1 and level-2 links, respectively, and $\mathbf{I}$ is the identity matrix (see Supplementary Note 6 for details). With such simple equations, we can readily derive the relationship among the joint angles for all the transition paths using the local Cartesian coordinate

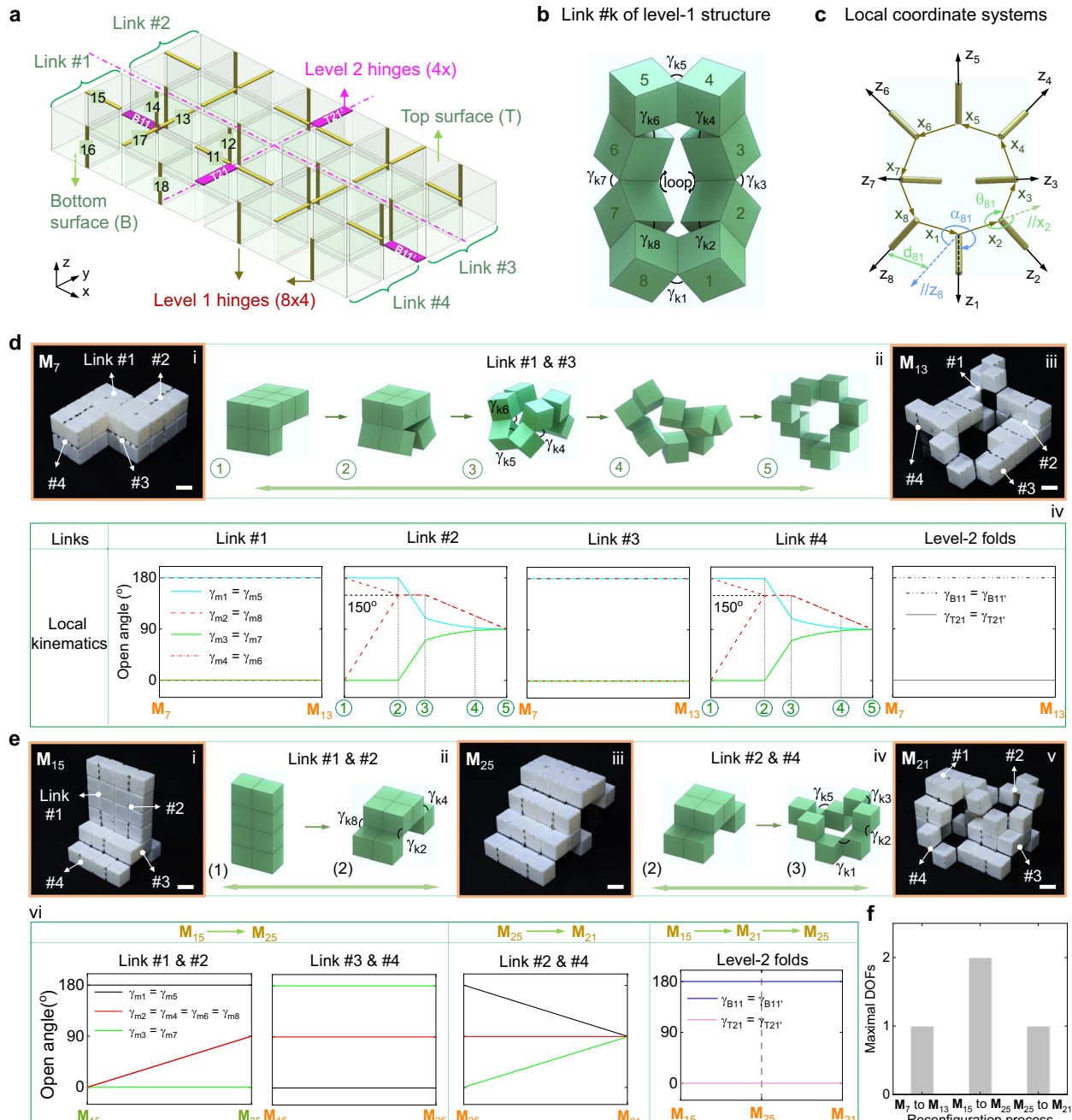

**Fig. 4 | Angle relationships during the shape morphing. a** Schematics of level-2 structures with labeled hinge connections on top and bottom surface. **b** Schematics of the opening angles $\gamma_{kj}$ ($k, j$ are integers with $1 \le k \le 4$ and $1 \le j \le 8$ denote the link and hinges opening angles, respectively) between adjacent cubes in level-1 structure. **c** Construction of eight local coordinate systems for the 8 hinges of level-1 structure. **d** The reconfiguration kinematics from node $M_7$ (i) to node $M_{13}$ (iii) in Fig. 3a, b: the involved shape-changing details of level-1 link #1 and #3 (ii) and

variations of the rotating angles for all folds (iv). Scale bar: 3 cm. **e** The reconfiguration kinematics from node $M_{15}$ (i) to node $M_{21}$ (v) by bypassing node $M_{25}$ (iii) in Fig. 3a, b: the involved shape-changing links #1 and #2 for the process from node $M_{15}$ to $M_{25}$ (ii) and links #2 and #4 for the process from node $M_{25}$ to $M_{21}$ (iv) and variations of all folds during these two processes (vi). Scale bar: 3 cm. **f** Low reconfiguration DOFs for the reconfiguration process in (**d**) (1 DOF) and (**e**) (1 or 2 DOF(s)).

systems presented in Fig. 4c (see Supplementary Note 7.1 for details).

To illustrate the simplicity of transition kinematics, we select two representative reconfiguration paths (node $M_7 \rightarrow$ node $M_{13}$ and node $M_{15} \rightarrow$ node $M_{21}$ in Fig. 3a) that transform from simple chain-like structures to complex architectures with ISLs ("Methods"). Figure 4d, e shows their detailed transition kinematics for these paths. It is observed that both shape-morphing paths involve only local and stepwise transition kinematics. For example, when transitioning from

node $M_7$ to $M_{13}$ (Fig. 4d, i and iii), only the joints in link #2 and #4 (Fig. 4d, i) are engaged in sequential rotations (Fig. 4d, ii), while the remaining joints in link #1, link #3, and level-2 joints remain stationary (Fig. 4d, iv) (see "Methods" for details). Similarly, the reconfiguration kinematics from node $M_{15}$ to $M_{21}$, bypassing node $M_{25}$, follows a straightforward linear angle relationship, as shown in Fig. 4e, i–vi. Despite these two reconfiguration processes representing the most complex shape morphing (see more details in Supplementary Figs. 17 and 18), they can be achieved using simple

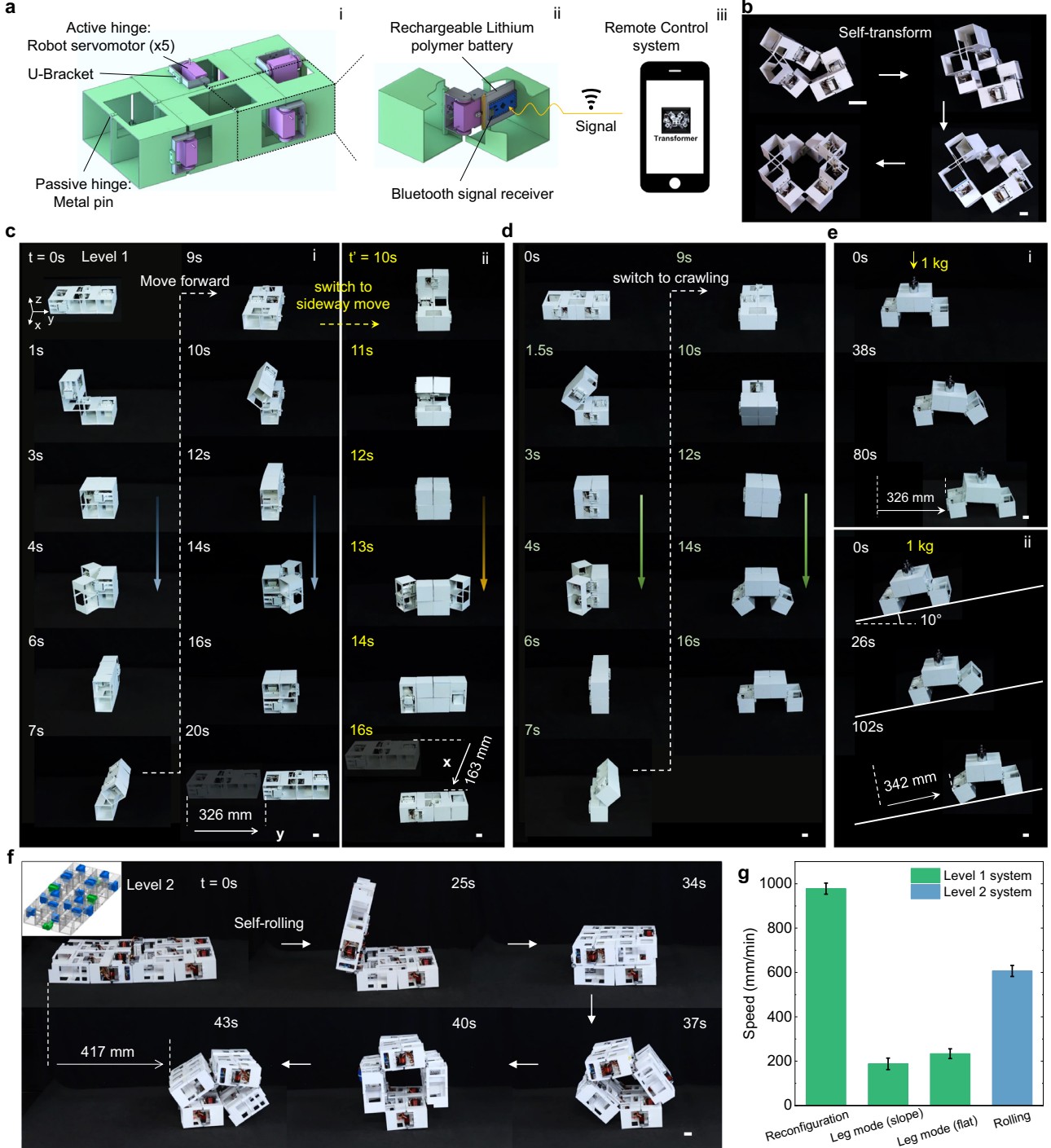

**Fig. 5 | Application in autonomous robotic transformer for multigait locomotion. a** Schematics of untethered actuation design details for the level 1 eight-cube-based structure: 5 electrically powered servomotors for active hinge rotation (i), onboard power system and Bluetooth wireless receiver to conduct reconfiguration order (ii) from customized remote control software (iii). **b** Demonstrated untethered shape morphing in the level-1 structure through looped mechanisms. **c**–**e** Shape transformation in level-1 structure for multigait locomotion. Scale bar: 3 cm. **c** Forward (i) and sideway locomotion (ii). **d** Locomotion gait switch from reconfiguration to legged walking; **e** Legged walking with carried payload on flat surface (i) and 10°-sloped surface (ii). **f** Demonstration of the specific positions of 22 active servomotors and the rolling locomotion of level-2 structure. Scale bar: 3 cm. **g** Locomotion speeds of both level-1 and level-2 structures in (**c**–**f**).

kinematics-based control. Moreover, Fig. 4f shows that the number of active DOFs for each step remains below 3 during these two reconfiguration processes, thanks to the specific looped interconnectivity of hierarchical structures. This is superior to previous designs, which either featured condensed[11,12,14,16,18,22,26] or completely discrete internal connections[25,27].

Given the unveiled simple transition kinematics of hierarchical structure and the low number of active DOFs during shape morphing, next, we explore and demonstrate their potential applications such as autonomous robotic transformers with adaptive locomotion, rapidly deployable self-reconfigurable architectures, and multifunctional space robots.

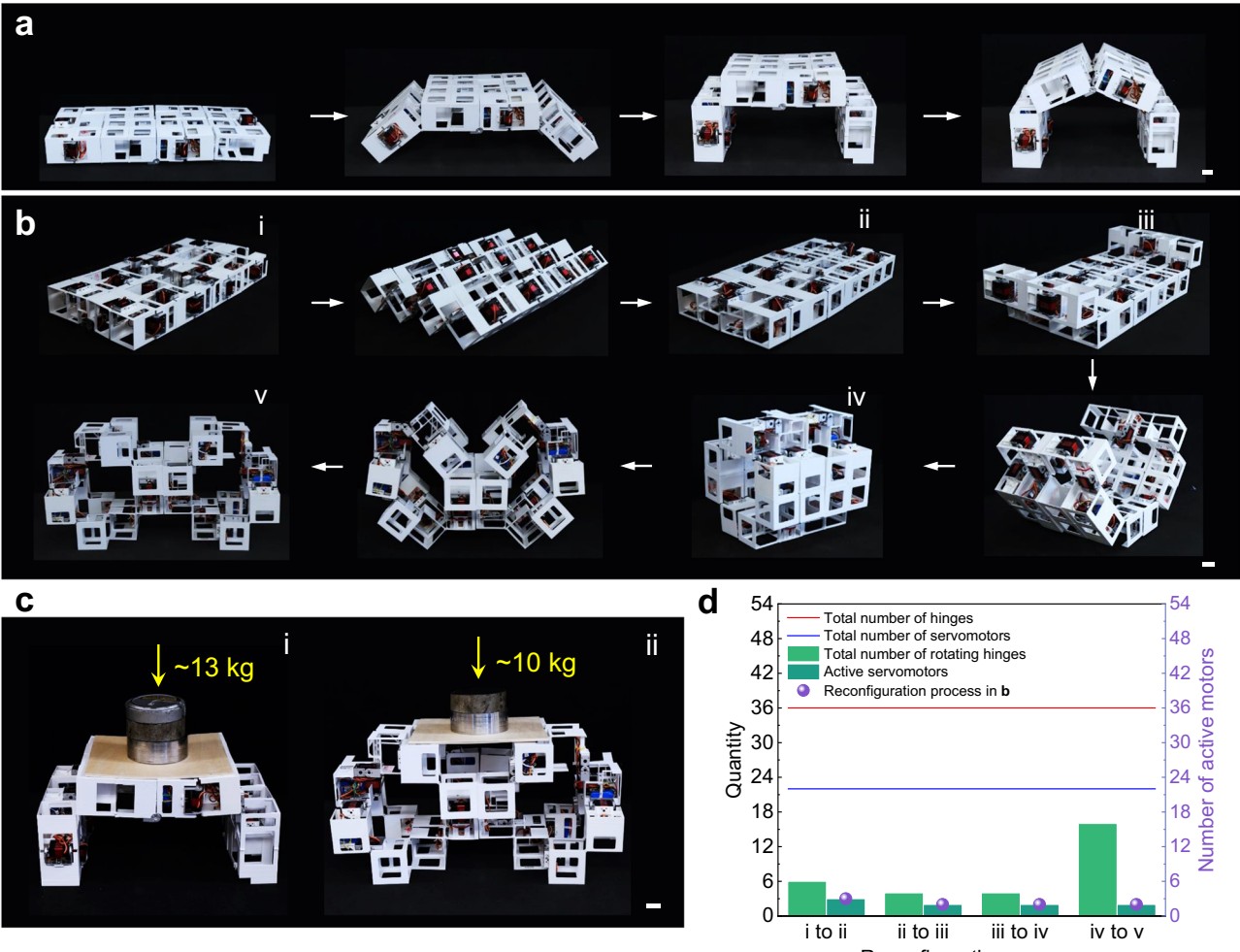

**Fig. 6 | Application of level-2 structures in self-deployable architectures.** Self-deployment into bridge and/or shelter-frame-like (**a**) and fully open 4-story building-like structures (**b**) with high loading capacity of over 10 kg (**c**). Scale bar: 3 cm. **d** Comparison among the total number of hinges, the total number of servomotors, the rotated hinges, and the actively actuated servomotors during the shape transformation shown in (**b**).

## Autonomous multigait robotic transformer

To achieve autonomous shape morphing in the hierarchical origami structure, we utilize servomotors to actuate the active joints, while passive joints are secured using metal pins (Fig. 5a, i). These servomotors are powered by onboard rechargeable batteries and controlled through a customized circuit board equipped with a Bluetooth signal receiver (Fig. 5a, ii, see more details in "Methods" and Supplementary Note 9). This setup enables untethered shape morphing via a developed remote control system (Fig. 5a, iii, see more details in Supplementary Figs. 19–21 and Supplementary Movies 4 and 8).

Thanks to the specific kinematics, even though there are a total number of 8 joints in a level-1 structure and 32 joints in a level-2 structure, only 5 (Fig. 5a) and 22 (Fig. 5f) servomotors are needed to accomplish all the reconfiguration paths in these structures (Fig. 5b–e in level 1, and Figs. 5f and 6a–c in level 2, respectively, see details in Supplementary Fig. 22). Importantly, the number of active servomotors involved in the reconfiguration paths does not exceed 3 (Fig. 6d). For the level-1 structure, it can rapidly and continuously transform from the compact planar state to 6R and 8R-looped linkage configurations via looped mechanisms within a few seconds (Fig. 5b and Supplementary Movie 4). Additionally, it can assume simple 2R chain-like configurations via chain-like mechanisms (Fig. 5c).

Next, we delve into harnessing active shape morphing for autonomous robotic multigait (Fig. 5c–e) and rolling (Fig. 5f)

locomotion. By following the chain-like reconfiguration loop path (Supplementary Fig. 7b), the level-1 structure can repeatedly transform its body shape to achieve impressive multigait robotic locomotion. For instance, it can perform forward or backward locomotion (one cycle is shown in Fig. 5c, i) at a rapid speed of approximately 1000 mm/min (3.07 body length/min) (Fig. 5g). Alternatively, it can change its movement direction from forward motion to sideway motion (Fig. 5c, ii) or switch its reconfiguration locomotion mode to a bipedal crawling mode (Fig. 5d and see more details in Supplementary Fig. 19c). Moreover, it is capable of carrying some payload (around 1 kg, equivalent to its self-weight) and climbing sloped surfaces (10°, Fig. 5e) at reduced speeds of approximately 225 mm/min and 190 mm/min (Fig. 5g), respectively. Furthermore, a similar chain-like reconfiguration allows us to demonstrate rolling-based mobility in the level-2 structure (Fig. 5f and Supplementary Movie 5) at a speed of about 600 mm/min (Fig. 5g).

## Rapidly deployable and scalable self-reconfigurable architectures

Moreover, the compact level-2 structure can effectively self-transform and rapidly deploy into architectural forms resembling bridges, tunnels, and shelters (Fig. 6a, b and Supplementary Movie 6), both with and without internal looped structures. This transformation occurs within 2 min, a notable advance compared to previous studies that required several hours and complex algorithms[11,23–26]. Additionally, it

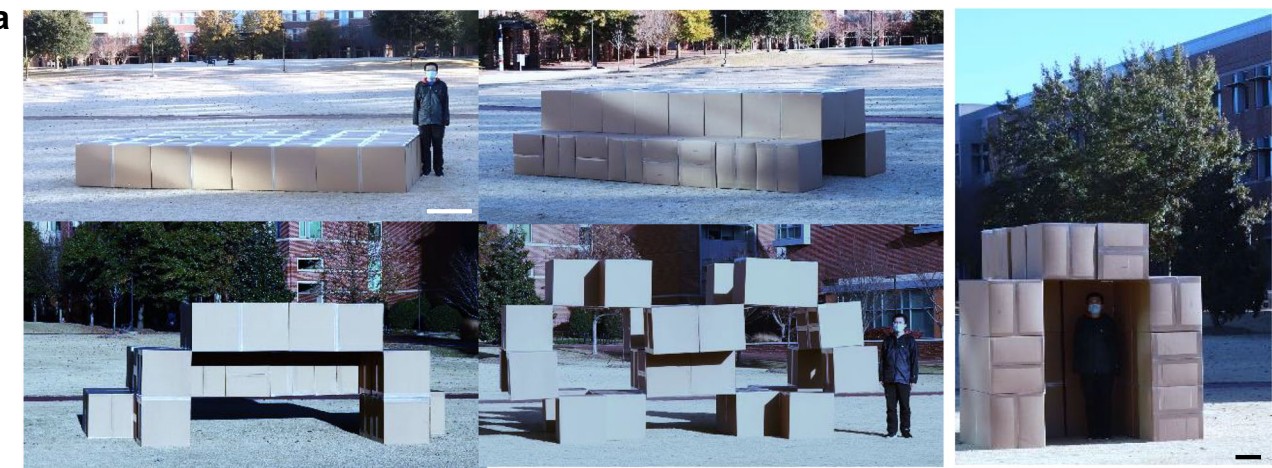

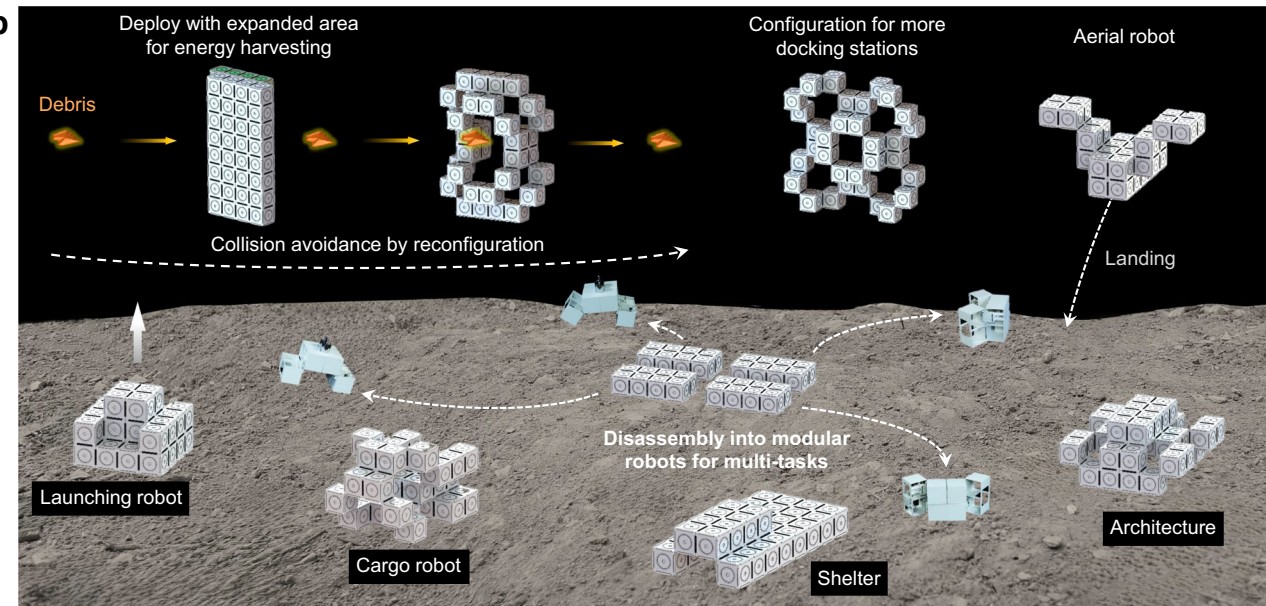

**Fig. 7 | Multifunctional applications of level-2 structures in scalable architectures and space robots. a** Meter-scale demonstration of deployable, shape-morphing architectures using cubic packaging boxes (side length of 60 cm). Scale bar: 30 cm. **b** Schematics of potential conceptual applications in versatile reconfigurable space robots and habitats.

can rapidly self-deploy into a fully open multi-story building-like structure, expanding its occupied volume fourfold (Fig. 6b, v). It can also quickly revert to a compact large cube (Fig. 6b, iv and Supplementary Movie 6). Due to its specific structural features (Supplementary Fig. 23, see more details in Supplementary Note 10), the reconfigured level-2 structure can bear substantial loads without collapsing, such as approximately 13 kg (over 3.5 times its self-weight) for the bridge- or tunnel-like structures and about 10 kg (over 2.5 times its self-weight) for the multi-story structure (Fig. 6c).

Notably, during the self-deployment from a compact planar structure to a complex multi-story open structure in Fig. 6b, the number of active motors remains low, never exceeding 3, despite the total number of 36 joints and 22 motors (Fig. 6d). For example, during the reconfiguration from the compact cube to the fully open structure (Fig. 6b, iv–v), only 2 active servomotors drive the rotation of 16 joints (Fig. 6d), demonstrating high reconfiguration efficiency.

As proof of concept, we demonstrate that these spatial hierarchical mechanism designs can be up-scaled to meter-sized buildings by assembling heavy-duty cardboard packing boxes (box side length 0.6 m). Starting from flat-packed cardboards with minimal space

requirements, they can be rapidly assembled for easy deployment and reconfiguration into various structurally stable meter-scale tunnels, shelters, and multi-story open structures (Fig. 7a and Supplementary Movie 7). Remarkably, the total volume occupied by the deployed multi-story open architecture is 200 times larger than the initial volume of the flat-packed cardboards (Supplementary Fig. 24). Collectively, these properties make the proposed design promising for potential applications as temporary emergency shelters and other autonomously rapidly deployable and reconfigurable temporary buildings.

## Discussion

The hierarchical and combinatorial designs in both the links and joints at multiple levels of hierarchical structures provide an extensive design space for creating various spatial looped folding patterns and architected origami-inspired structures capable of shape morphing. It creates hierarchical origami-based metamaterials with (1) fewer active reconfiguration mobilities, (2) simple reconfiguration kinematics to facilitate practical control and actuation, and (3) rich shape-morphing capability adaptable to various applications. The hierarchical architecture couples the closed-loop mechanisms within and across each

hierarchical level. Despite the large number of joints involved, the hierarchical looped mechanisms inherently impose geometric constraints that dramatically reduce the number of active DOFs required for shape morphing. This reduction greatly simplifies both actuation and control without sacrificing rich shape-morphing capability, which previously required the actuation of each DOF individually in reconfigurable origami metamaterials and robots. It also enables the feasibility of inverse designs, allowing for imitating target shapes and structures (Supplementary Figs. 20, 25 and 26, see theoretical details in Supplementary Note 11).

Our design strategy combines structural hierarchy with overconstrained looped kinematic mechanism without considering elastic deformation in the hinges and cubes. Practically, the elastic deformation or slack, especially in the hinges, could cause the system to be floppy or potentially deviate from the desired non-bifurcated and/or bifurcated kinematic paths. As demonstrated in the multimaterial 3D-printed level-2 structure in Fig. 3a, the soft hinges are printed thin with little stiffness to ensure almost free rotation. Thus, in addition to bending for rotation motion, the hinges also undergo certain twisting deformation, potentially causing the structure to deviate from their ideal kinematic paths. However, deviations occur only during the complex reconfiguration processes, e.g., from configuration $M_7$ to $M_{13}$ in Fig. 3a. Such deviations are suppressed when the reconfiguring structure exhibits structural symmetries, e.g., from configuration $M_D$ to Configuration $M_E$ in Fig. 3a preserving $x$-$y$ and $z$-$y$ plane symmetries. The slack can be avoided by fabricating hinges with a low ratio of bending stiffness to twisting stiffness. This will help to suppress its twisting deformation to follow the kinematic paths without making the structure overly floppy. For systems fabricated with high-precision rigid links and hinges, slack or elastic deformation can be minimized or eliminated, as demonstrated in the prototype of both level-1 and level-2 structures with 3D-printed rigid cubes and rigidly rotatable hinges in Fig. 5a. Similar to studied 2D rigidly foldable origami structures, the reconfiguration kinematics of the system becomes energy scale independent. Thus, the system can rigorously follow its bifurcated reconfiguration kinematic path via fewer number of actuation hinges to smoothly reconfigure into all desired configurations without any locking issues as demonstrated in Figs. 5 and 6.

We note that there are several limitations of this work. First, the load bearing capacity of some reconfigured 3D architectures is still limited, which could hinder their practical engineering and structural applications, especially at meter scales. The load bearing capacity is dependent of not only the transformed architectures (see the free body diagrams of force analysis for example in Supplementary Fig. 23), but also the bending stiffness of both cubes and hinges and the structural designs of the hinges. The hinges are imitated with 3D-printed soft rubber-like materials or tapes with low bending rigidity that facilitate the bending and rotation motion but sacrifice the load-carrying capabilities. The load bearing capacity could be improved by using stronger materials with high bending rigidity or locking hinges or devices at either 90° or 180° folded angles. Second, the shape-morphing capability for robotic applications is limited to multi-gait motion demonstrated in this work. How to leverage the rich shape-morphing capability for diverse and adaptive robotic locomotion in unstructured environments remains to be uncovered. Third, the demonstration of self-deployment and self-reconfiguration is limited to centimeter-scale prototypes while the meter-scale demo is done manually due to the limitation of both power and servomotors. At large scales, the heavier self-weight of cubes cannot be neglected, which requires high-torque servomotors and high-power batteries to generate sufficient torque output to counter the gravity and drive the folding.

Moving forward, these limitations also open new opportunities for future researches in morphing matter. First, this work explores only a small region of the tremendous design space in morphing matter to showcase its potential. The vast combinatorial folding patterns arise from the combinatorial connections in the base units, as well as within and across each hierarchical mechanism (Supplementary Fig. 3). These combinatorial hierarchical mechanisms are generalizable and can be applied to construct similar reconfigurable hierarchical metastructures composed of any shape-morphing spatial closed-loop mechanism for easy actuation and control yet rich shape morphing. For example, the cube units can be replaced by other composed geometrical shapes, such as thick plates with substantially reduced thickness dimension, tetrahedrons, and triangular-shaped prisms, or extended to genuine volumetric 3D structures (examples are provided in Supplementary Figs. 27 and 28, with more details in Supplementary Note 12).

Second, this work focuses on exploring the reconfiguration kinematics of the hierarchical origami systems by modeling the system as idealized hierarchical rigid mechanisms and neglecting the deformation in both the cubes and hinges. However, in scenarios when such elastic deformation are non-negligible, similar to the non-rigidly deformable origami metamaterials in origami engineering, the over-constrained looped kinematic mechanisms become energy scale dependent, considering the potentially involved complex deformation in the cubes, hinges, and architectures during reconfiguration such as bending, stretching, twisting, and shearing or combined. Consequently, it will transform the rigid mechanisms into both reconfigurable and deformable architected materials and structures, which couples kinematics with mechanics. Such coupling will enrich new kinematics, mechanics, transformed configurations, reconfiguration paths, and reprogrammable mechanical behaviors such as multistability and stiffness anisotropy. Specially, how the energy scale affects the kinematic bifurcated paths and how the coupled kinematic bifurcation and elasticity change both the reconfigurations and mechanical responses of bifurcated mechanical metamaterials remain to be uncovered. We envision such studies could also find broad applications in reprogrammable mechanical computing, mechanical memory, and mechanical metamaterials.

Third, considering these multi-capabilities in conjunction with scalability, modularity, and disassemblability, we envision diverse applications in robotics, architecture, and even in space. Figure 7b conceptually illustrates potential applications in multitask adaptive shape-morphing space robots and habitat (Supplementary Movie 8). The hierarchical origami architectures could be deployed with largely increased exposed surface areas for enhanced solar energy harvesting, and reconfigured to avoid debris collision or accommodate more docking stations. It could also serve as reconfigurable space habitat or be des-assembled into modular robots for multitask exploration. For large-sized structures, the feasibility of actuation in a space environment is considerably higher, primarily due to the absence of gravity and the absence of ground-based collisions that can impede complex shape-morphing processes on Earth.

## Methods

### Sample fabrication of cube-based origami structures

To demonstrate the shape morphing in cube-based origami structures, we used two ways to fabricate and assemble the hollow cubes. One is for quick shape-morphing demonstration by directly 3D printing individual cubes with cube size of 2 cm (Stratasys Connex Objet-260 with stiff materials of Vero PureWhite) and connecting them with adhesive plastic tapes (Scotch Magic Tape, 6122) as free-rotation hinges (Supplementary Figs. 5, 8 and 11–13). The other is for easy assembly and disassembly demonstration by 3D printing Lego-like pieces of thin rigid plates (Fig. 3). Two types of thin plates were printed (Supplementary Fig. 7a): one is a thin rigid plate with interlocking teeth (Vero PureWhite) for assembling into a hollow cube,

the other is a connection piece composed of two connected thin rigid plates with soft hinges made of rubber-like materials (Agilus-black) through 3D multimaterial printing. The connection piece is used to connect two neighboring cubes at any selected hinge locations with the soft hinges facilitating the free rotation of cubes. The cube size is 3 cm.

## Fabrication of autonomous robotic transformers

The cubes were 3D-printed with ABS printing materials (QIDI Tech X-Max 3D printer) with cube size of 81.5 mm and mass of 40 g. To ensure the compact contacts between the 3D-printed cube components, we created open areas at the joints positions and use the U-shaped bracket to hold electronic elements (Fig. 5a, i). Each motor (DSservo RDS3225) was powered by a 3.7 V LiPo battery and controlled via its specific control board (Adafruit ItsyBitsy nRF52840 Express). Additional chips were incorporated for accommodating the JST connector for the battery (Adafruit Pro Trinket LiIon/LiPoly Backpack Add-On) and for adaption of the supply voltage (SparkFun Logic Level Converter−Bi-Directional). The control boards were identified by a numeric ID and communicated with each other via Bluetooth by following a serial framework, where each controller receives the information from the previous one and sends them to the next one. More details can be found in Section S10 of Supplementary Information.

## Fabrication of meter-scale samples

The cubes used in the meter-scale shape-morphing architectures in Fig. 7a were heavy-duty cardboard packing boxes (Recycled Shipping Box, Kraft) with dimensions of 0.6 m × 0.6 m × 0.6 m. Boxes were connected using the fiber-reinforced ultra-adhesive tape (BOMEI PACK Transparent Bi-Directional Filament Strapping Tape).

## Fundamental principles governing the shape transformation

Given the mechanical kinematic mechanism's structural features, the core of the shape transformations in these structures involves changes in the spatial positions of specific structural elements resulting from the directional rotations of internal hinges and their interconnections. Technically[10,22,51], this operation can be mathematically modeled using a rotation matrix $\mathbf{t}$ (see details in Supplementary Note 4.2). Thus, the shape transformations of any leveled structures can be denoted as:

$$\mathbf{M}' = \mathbf{t}\mathbf{M} \tag{1}$$

where $\mathbf{M}'$ represents the transformed shape from shape $\mathbf{M}$, with both $\mathbf{M}$ and $\mathbf{M}'$ reflected in Fig. 3a and Supplementary Fig. 9 as $\mathbf{M}_k$, and $\mathbf{t}$ represents the mathematical operations between them. In our analysis, we initially build a fixed global Cartesian coordinate system at the bottom center of the original shape (see the inset at the initial shape in

morphing process. Mathematically, we can thus determine the new positions of the rotated cubes as follows:

$$\boldsymbol{v}_{n\_new} = \mathbf{t}_n \boldsymbol{v}_{n\_local} + \boldsymbol{d}_n \tag{2}$$

where $\boldsymbol{v}_{n\_local}$ and $\boldsymbol{v}_{n\_new}$ represent the body center vectors of cube #$n$ before and after shape morphing, respectively, in the local and fixed global coordinate systems. $\mathbf{t}_n$ is a general functional form including all directional rotations of cube #$n$ (Supplementary Fig. 10b−e), see the systematic analytical details in Supplementary Note 4.2. $\boldsymbol{d}_n$ is the translational vector between the fixed global coordinate system and the local coordinate system of cube #$n$. Note that all shape matrices of the initial and reconfigured shapes are described in the fixed global coordinate systems.

We validate the theoretical framework by modeling the shape-morphing process of reconfiguration loop 1, i.e., from the initial shape $M_A$ to shape $M_F$, passing through shapes $M_B$, $M_C$, $M_D$ and $M_E$. The shape matrix of the initial shape is determined first in the fixed global coordinate system. Based on Eqs. (1) and (2), we rationally derive the new positions of the rotated cubes in new shapes accordingly. Specifically, we analyze the process from morphing from shape $M_D$ to shape $M_E$, where a total of 24 cubes are involved. To provide a representative example, we select cube #20 and theoretically derive its new spatial positions. Subsequently, we compare these derived solutions with experimental results to validate our proposed theoretical framework.

Starting from the initial shape $M_A$, we derive the shape matrix of $M_D$, as expressed in Eq. (3). Consequently, in the fixed global coordinate system, we obtain the explicit spatial vector for cube #20 $\boldsymbol{v}_{20}^{M_D}$ with $\boldsymbol{v}_{20}^{M_D} = (1,3,1)^T$. During the shape-morphing process, cube #20 undergoes rotation along the $x$-axis within the locally built coordinate systems (Supplementary Fig. 10a, iv). To derive its new positions, we first calculate its spatial vector in the local coordinate systems, represented by $\boldsymbol{v}_{20\_local}^{M_D} = (1,1,1)^T$. Utilizing a 90° $x$-directional rotation, we then derive its new coordinates in the global coordinate systems using Eq. (2) with explicit derivation details as:

$$\boldsymbol{v}_{30\_new}^{M_E} = \begin{bmatrix} 1 & 0 & 0 \\ 0 & \cos(90°) & \sin(90°) \\ 0 & -\sin(90°) & \cos(90°) \end{bmatrix} \begin{pmatrix} 1 \\ 1 \\ 1 \end{pmatrix} + \begin{pmatrix} 0 \\ 2 \\ 0 \end{pmatrix} = (1,3,-1)^T \tag{3}$$

Within the built fixed global coordinate systems, we extract the experimental result pertaining to the spatial position of cube #20 in the global coordinate system, denoted as $\boldsymbol{v}_{30}^{M_E} = (1,3,-1)^T$. The theoretical model is in excellent agreement with the experimental result. In order to derive the shape matrices of shapes $M_D$ and $M_E$ in reconfiguration loop 1, we need firstly determine the shape matrix of the

$$\mathbf{M_A} = (\boldsymbol{v}_1, \boldsymbol{v}_2, \boldsymbol{v}_3, \cdots, \boldsymbol{v}_{32})$$

$$= \begin{pmatrix} -7 & -7 & -7 & -7 & -5 & -5 & -5 & -5 & -3 & -3 & -3 & -3 & -1 & -1 & -1 & -1 \\ (-3), & (-1), & (1), & (3), & (3), & (1), & (-1), & (-3), & (-3), & (-1), & (1), & (3), & (3), & (1), & (-1), & (-3), \\ 1 & 1 & 1 & 1 & 1 & 1 & 1 & 1 & 1 & 1 & 1 & 1 & 1 & 1 & 1 & 1 \\ 1 & 1 & 1 & 1 & 3 & 3 & 3 & 3 & 5 & 5 & 5 & 5 & 7 & 7 & 7 & 7 \\ (-3), & (-1), & (1), & (3), & (3), & (1), & (-1), & (-3), & (-3), & (-1), & (1), & (3), & (3), & (1), & (-1), & (-3), \\ 1 & 1 & 1 & 1 & 1 & 1 & 1 & 1 & 1 & 1 & 1 & 1 & 1 & 1 & 1 & 1 \end{pmatrix} \tag{4}$$

Fig. 3a, ii). Subsequently, we construct a local coordinate system at each fold to derive the body center coordinates of the rotated cube structural components (Supplementary Fig. 10a, b) in each shape-

initial shape $M_A$, which is presented with explicit components as:

Then, combining Eqs. (1)−(4), we can finally obtain the shape $M_D$ and shape $M_E$ as:

$$\mathbf{M_D} = (\boldsymbol{v}_1, \boldsymbol{v}_2, \boldsymbol{v}_3, \cdots, \boldsymbol{v}_{32})$$

$$= \left( \begin{pmatrix} -1 \\ -3 \\ 7 \end{pmatrix}, \begin{pmatrix} -1 \\ -1 \\ 7 \end{pmatrix}, \begin{pmatrix} -1 \\ 1 \\ 7 \end{pmatrix}, \begin{pmatrix} -1 \\ 3 \\ 7 \end{pmatrix}, \begin{pmatrix} -3 \\ 3 \\ 5 \end{pmatrix}, \begin{pmatrix} -3 \\ 1 \\ 5 \end{pmatrix}, \begin{pmatrix} -3 \\ -1 \\ 5 \end{pmatrix}, \begin{pmatrix} -3 \\ -3 \\ 5 \end{pmatrix}, \begin{pmatrix} -3 \\ -3 \\ 3 \end{pmatrix}, \begin{pmatrix} -3 \\ -1 \\ 3 \end{pmatrix}, \begin{pmatrix} -3 \\ 1 \\ 3 \end{pmatrix}, \begin{pmatrix} -3 \\ 3 \\ 3 \end{pmatrix}, \begin{pmatrix} -1 \\ 3 \\ 1 \end{pmatrix}, \begin{pmatrix} -1 \\ 1 \\ 1 \end{pmatrix}, \begin{pmatrix} -1 \\ -1 \\ 1 \end{pmatrix}, \begin{pmatrix} -1 \\ -3 \\ 1 \end{pmatrix}, \right.$$
$$\left. \begin{pmatrix} 1 \\ -3 \\ 1 \end{pmatrix}, \begin{pmatrix} 1 \\ -1 \\ 1 \end{pmatrix}, \begin{pmatrix} 1 \\ 1 \\ 1 \end{pmatrix}, \begin{pmatrix} 1 \\ 3 \\ 1 \end{pmatrix}, \begin{pmatrix} 3 \\ 3 \\ 3 \end{pmatrix}, \begin{pmatrix} 3 \\ 1 \\ 3 \end{pmatrix}, \begin{pmatrix} 3 \\ -1 \\ 3 \end{pmatrix}, \begin{pmatrix} 3 \\ -3 \\ 3 \end{pmatrix}, \begin{pmatrix} 3 \\ -3 \\ 5 \end{pmatrix}, \begin{pmatrix} 3 \\ -1 \\ 5 \end{pmatrix}, \begin{pmatrix} 3 \\ 1 \\ 5 \end{pmatrix}, \begin{pmatrix} 3 \\ 3 \\ 5 \end{pmatrix}, \begin{pmatrix} 1 \\ 3 \\ 7 \end{pmatrix}, \begin{pmatrix} 1 \\ 1 \\ 7 \end{pmatrix}, \begin{pmatrix} 1 \\ -1 \\ 7 \end{pmatrix}, \begin{pmatrix} 1 \\ -3 \\ 7 \end{pmatrix} \right)$$

$$\mathbf{M_E} = (\boldsymbol{v}_1, \boldsymbol{v}_2, \boldsymbol{v}_3, \cdots, \boldsymbol{v}_{32})$$

$$= \left( \begin{pmatrix} -1 \\ -7 \\ 3 \end{pmatrix}, \begin{pmatrix} -1 \\ -5 \\ 5 \end{pmatrix}, \begin{pmatrix} -1 \\ 5 \\ 5 \end{pmatrix}, \begin{pmatrix} -1 \\ 7 \\ 3 \end{pmatrix}, \begin{pmatrix} -3 \\ 7 \\ -1 \end{pmatrix}, \begin{pmatrix} -3 \\ 3 \\ 5 \end{pmatrix}, \begin{pmatrix} -3 \\ -3 \\ 5 \end{pmatrix}, \begin{pmatrix} -3 \\ -7 \\ -1 \end{pmatrix}, \begin{pmatrix} -3 \\ -5 \\ -1 \end{pmatrix}, \begin{pmatrix} -3 \\ -1 \\ 3 \end{pmatrix}, \begin{pmatrix} -3 \\ 1 \\ 3 \end{pmatrix}, \begin{pmatrix} -3 \\ 5 \\ -1 \end{pmatrix}, \begin{pmatrix} -1 \\ 3 \\ -1 \end{pmatrix}, \begin{pmatrix} -1 \\ 1 \\ 1 \end{pmatrix}, \begin{pmatrix} -1 \\ -1 \\ 1 \end{pmatrix}, \begin{pmatrix} -1 \\ -3 \\ -1 \end{pmatrix}, \right.$$
$$\left. \begin{pmatrix} 1 \\ -3 \\ -1 \end{pmatrix}, \begin{pmatrix} 1 \\ -1 \\ 1 \end{pmatrix}, \begin{pmatrix} 1 \\ 1 \\ 1 \end{pmatrix}, \begin{pmatrix} 1 \\ 3 \\ -1 \end{pmatrix}, \begin{pmatrix} 3 \\ 5 \\ -1 \end{pmatrix}, \begin{pmatrix} 3 \\ 1 \\ 3 \end{pmatrix}, \begin{pmatrix} 3 \\ -1 \\ 3 \end{pmatrix}, \begin{pmatrix} 3 \\ -5 \\ -1 \end{pmatrix}, \begin{pmatrix} 3 \\ -7 \\ 1 \end{pmatrix}, \begin{pmatrix} 3 \\ -3 \\ 5 \end{pmatrix}, \begin{pmatrix} 3 \\ 3 \\ 5 \end{pmatrix}, \begin{pmatrix} 3 \\ 7 \\ 1 \end{pmatrix}, \begin{pmatrix} 1 \\ 7 \\ 3 \end{pmatrix}, \begin{pmatrix} 1 \\ 5 \\ 5 \end{pmatrix}, \begin{pmatrix} 1 \\ -5 \\ 5 \end{pmatrix}, \begin{pmatrix} 1 \\ -7 \\ 3 \end{pmatrix} \right)$$

(5)

## Reconfiguration kinematics in Fig. 4

The following gives the reconfiguration kinematic details for the morphing process shown in Fig. 4d, e. Specially, we label the opening angles of four level-1 link structure as $\gamma_{km}$ ($k$ and $m$ are integers with $1 \le k \le 4$ as the $k$th link while $1 \le m \le 8$ as the $m$th rotating folds between two adjacent cubes $m$ and $m+1$ ($m+1 \to 1$ when $m=8$), see details in Fig. 4a, b), and the level-2 folds angles separately as $\gamma_{B1I}$, $\gamma_{B1I'}$, $\gamma_{T2I}$ and $\gamma_{T2I'}$ (Fig. 4a, b and B and T represent the bottom and top surfaces, respectively).

The selected two reconfiguration processes exhibit only local and stepwise transition kinematics. From shape $M_7$ to $M_{13}$ (Fig. 4d, i and iii), only the folds of links #2 and #4 (Fig. 4d, i) are involved with sequential rotations (Fig. 4d, ii) and the remaining folds of link #1, link #3 and the level-2 keep unchanged (Fig. 4d, iv). For kinematic details of the reconfigured links #2 and #4 shown in Fig. 4d, iv, during the initial process ① → ②, we only need to linearly change the folds angle $\gamma_{m4,6}$ ($m=2$ or $4$) from 180° to $\gamma_0$ (Here we set $\gamma_0$ as 150° while it ranges from 90° to 180°; see more details in Supplementary Fig. 17) and meanwhile linearly increase $\gamma_{m2,8}$ from 0° to $\gamma_0$. Then, in the following process ② → ③, we can maintain folds angles $\gamma_{m2,4,6,8}$ as $\gamma_0$ while both linearly decreasing $\gamma_{m1,5}$ from 180° to $\sin^{-1}[(\sin \gamma_0)^2/(1+(\cos \gamma_0)^2]$ ($\approx 109.5°$ for $\gamma_0 = 150°$) and augmenting $\gamma_{m3,7}$ from 0° to $180° - \sin^{-1}[(\sin \gamma_0)^2/(1+(\cos \gamma_0)^2]$ ($\approx 70.5°$ for $\gamma_0 = 150°$). Lastly, for ③ → ④ → ⑤, we can simultaneously transfigure links #2 and #4 as 8R-looped rigid linkage with kinematics as $\gamma_{m1,5} = \sin^{-1}[(\sin \gamma_{m2})^2/(1+(\cos \gamma_{m2})^2]$, $\gamma_{m3,7} = 180° - \sin^{-1}[(\sin \gamma_{m2})^2/(1+(\cos \gamma_{m2})^2]$ ($\gamma_{m2}$ reducing from $\gamma_0$ to 90°) while $\gamma_{m2} = \gamma_{m4} = \gamma_{m6} = \gamma_{m8}$ (see Supplementary Note 6) to reach shape $M_{13}$. Moreover, as illustrated in Fig. 4e, vi that displays sequential and local kinematic features, we note that the reconfiguration kinematics from shape $M_{15}$ to $M_{21}$ by bypassing shape $M_{25}$ (Fig. 4e, i–v) are much simpler with only linear angle relationships.

## Inverse design to imitate target shapes

Inverse design to imitate target shapes for special application scenarios can also be accessible for our hierarchical structures. However, the imitating process of our inverse design is different from previous designs by presetting material/structural patterns to purposely retain the target shapes. Our inverse design method is based on the selection algorithm from the reconfigured shape library by following several steps.

First is to build a database for the configuration library. Each cube can be treated as a spatial voxelated pixel with its geometrical center represented by a vector. Then, we can use a matrix to characterize a morphed shape, where the spatial positions of composed cubes are described by their corresponding vectors. For example, for all the combinatorially designed level-2 structures shown in Supplementary Fig. 5, for one special design $k$, all its reconfigured shapes can be summarized into:

$$(\mathbf{M}_{k1}, \cdots, \mathbf{M}_{kn}) \tag{6}$$

where $\mathbf{M}_{kn}$ represent the mathematically expressed forms of the $n$th reconfigured shapes in the transition tree for the $k$th combinatorically designed level-2 structures.

Second is to compose all the combinatorically designed level-2 structures into the database matrix $\mathbf{D}$ in the form of:

$$\mathbf{D} = \begin{pmatrix} \mathbf{M}_{11} & \mathbf{M}_{12} & \cdots & \mathbf{M}_{1i} & 0 & 0 & 0 & 0 & 0 & 0 \\ \mathbf{M}_{21} & \mathbf{M}_{22} & \cdots & \mathbf{M}_{2i} & \cdots & \mathbf{M}_{2j} & 0 & 0 & 0 & 0 \\ \vdots & \vdots & \vdots & \vdots & \vdots & \vdots & \vdots & \vdots & \vdots & \vdots \\ \mathbf{M}_{(k-1)1} & \mathbf{M}_{(k-1)2} & \cdots & \mathbf{M}_{(k-1)i} & \cdots & \mathbf{M}_{(k-1)j} & \cdots & \mathbf{M}_{(k-1)m} & 0 & 0 \\ \mathbf{M}_{k1} & \mathbf{M}_{k2} & \cdots & \mathbf{M}_{ki} & \cdots & \mathbf{M}_{kj} & \cdots & \mathbf{M}_{km} & \cdots & \mathbf{M}_{kz} \end{pmatrix} \tag{7}$$

where $z$ stands for the maximum number of reconfigured shapes by the $k$th level-2 structure.

Third is to discretize the target shape into cube-shaped voxelated pixels and mathematically convert it into a mathematical matrix $\mathbf{T}$.

Last is to find the shapes in the database that match for the target shape by comparing the matrix $\mathbf{T}$ with the components of database matrix, i.e., $\mathbf{D}_{ij}$. There are two criterions to find out the optimal imitated shape: (1) find the smallest value of the error function **Errf** defined as:

$$\mathbf{Errf} = \| \mathbf{T} - \mathbf{D}_{ij} \| / \| \mathbf{T} - \mathbf{D}_{ij} \|_{\max} \tag{8}$$

wherein $\| \ \|$ represent the mode of matrix and usually $\| \mathbf{T} - \mathbf{D}_{ij} \|_{\max}$ is determined as $\| \mathbf{T} \| + \| \mathbf{D}_{ij} \|_{\max}$ for simplicity. (2) The conditions that guarantee the imitated shapes whose cube pixels are with approximately the same absolute spatial positions with the target shape, i.e.:

$$\| \boldsymbol{v}_\mathbf{T} - \boldsymbol{v}_{\mathbf{D}_{ij}} \| = 0 \tag{9}$$

Finally, we can obtain the most approximately imitated shape $\mathbf{M}_{km}$ from the database. The inverse design method is briefly summarized in Supplementary Fig. 25.

## Simulation by customized software

A model has been developed for simulation in ROS-Gazebo (Supplementary Figs. 20 and 21 and Supplementary Movies 5 and 8). For simplicity, a single design composed of the four lateral faces of a cube is used to model every module of the robot. The connections between

the modules are modeled as revolute joints (either passive or actuated). Additional blocks are used to replicate the positions and masses of the motors in the real system. Kinematic constraints are implemented to model the robot as a closed kinematic chain.

## Data availability
The authors declare that the data supporting the findings of this study are available within the article and its Supplementary Information files. Source data are provided with this paper.

## Code availability
The code used for the analyses is deposited via Zenodo at https://doi.org/10.5281/zenodo.12690922.

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

## Acknowledgements

J.Y. acknowledges the funding support from NSF (CMMI-2005374 and CMMI-2126072). H.S. acknowledges the funding support from NSF 2231419. The authors acknowledge the helpful discussions with Dr. K. Bertoldi and Dr. M. Yim.

## Author contributions

Y.L. and J.Y. proposed the idea. Y.L. conducted theoretical and numerical calculations. Y.L. and Y.C. designed and performed experiments on shape-morphing prototypes. A.D. and J.Z. designed and performed experiments on untethered actuation of shape-morphing prototypes. Y.L., A.D., H.S. and J.Y. wrote the paper. H.S. and J.Y. supervised the research. All the authors contributed to the discussion, data analysis, and editing of the manuscript.

## Competing interests

The authors declare no competing interests.
