## [Peer Review File · Nature Communications]

Adaptive hierarchical origami-based metastructuresREVIEWER COMMENTS

Reviewer #1 (Remarks to the Author):

First, this reviewer would like to apologize for the long delay in providing a review. There were some personal matters that interfered with being able to provide a timely evaluation of this paper. This paper presents a concept for hierarchical shape morphing with block-like kirigami structures. The paper is well written, the research is thoroughly performed, and the work presents novel ideas worthy of publication.

First, a comment regarding the overlap with the authors' previous work presented in reference #36 – there is some overlap with the previous work in terms of concepts, methods, capabilities, and overall execution. The systems are physically similar, have a large range of shape morphing geometries and require few DOFs to reconfigure. This overlap is not ideal especially when attempting to publish this work in a premier level journal such as Nature Communications. With that said however, this reviewer does not see this overlap as a deal-breaker, and in fact the new paper is in many ways distinct and novel compared to the previous publication. The hierarchical concept for arranging and reconfiguring the systems is newly presented here and is quite different from the ad-hoc approach presented previously. Furthermore, there is now a more systematic approach for the design of the systems through hierarchical linkages. This paper also presents novel concepts for locomotion of the block like systems. These aspects are likely sufficient to justify the work to be published, however the authors should consider the following comments which could further strengthen and distinguish the paper.

1 First, the paper needs to take a more balanced and measured approach, especially since this is not the first paper with the theme of block-like morphing matter. It would be helpful if the paper discusses limitations of the work, and discuss areas for future work --- This is an exciting area for morphing matter, so what's next? --- For example, the discussion of using tetrahedrons, prisms and other shapes is done exceptionally well, and other current frontiers could be discussed (perhaps without requiring a study that is in so much in depth). The next four comments, all somewhat related, could perhaps be highlighted as challenges/limitations/opportunities – some could be explored here, or they could be discussed as topics for future study. Other discussions of limitations and future work would also be appreciated.

2 The paper uses a concept of what are technically over-constrained linkages. In practice, if such linkages are fabricated with high precision and with rigid links and hinges, then they become highly susceptible to locking and not following their kinematics properly (e.g. due to small imperfections). In contrast most such system are instead fabricated with some 'slack' or elastic deformation built in which allows the systems to move more freely. The issue with this slack is that it can lead to rather floppy systems with little stiffness. From the videos it appears that the systems here are indeed fabricated with some slack, and are somewhat floppy. Can the authors discuss this slack in more detail? Are there issues with the systems following their kinematic paths? Are the systems sometimes overly floppy leading to unwanted deformations? Is there some systematic guidance that they can give related to this?

3 Highly related to #2 above, the systems here, much like origami/kirigami rely on bifurcations to go into the desired kinematic paths. This can be especially challenging and tricky with complex systems. For example 'Self-folding origami at any energy scale' by Pinson et al. in Nature Communication, and other papers in the field have discussed these issues in some detail. Can the practicality of entering a desired path be briefly explored here? Are the systems always reliable, or are there issues with entering some of the potentially higher energy modes? A brief discussion related to this issue would help.

4 Again, related to the 2&3; but with a slightly different direction: The work here is focused purely on the kinematics of the systems. This focus is fine for now; however, the fields of origami/kirigami has moved to performing much more advanced mechanics based simulations. These simulations take into account more of the physics of the model including stiffness and deformations, and are well suited to dealing with some of the challenges in 2&3 above, and more. Furthermore, mechanics based models can give substantially more insight on physical behaviors, performance, and applications. Using such models to study these systems is a natural next step related to this work. Again, maybe this is something that does not need to be done here, but could be discussed as future work.

5 The stiffness and load bearing capability of the systems here is not quite that impressive compared to other origami/kirigami systems. The authors should distinctly highlight that as a current limitation. Perhaps more detailed modeling as in (4) together with the vast array of possible geometries can offer some unique geometries that come closer to other origami/kirigami designs.

It is likely alright if the topics in 2-5 above are not fully explored/addressed, however they should be discussed at least in some detail in this paper.

Reviewer #2 (Remarks to the Author):

The paper presents a method that uses the combinatorial pairing of edges between adjacent cubes to construct rigid-foldable origami that can morph into multiple reconfigurations. The strategy (denoted as hierarchical) explores the permutations that exist for connecting edges of neighboring cubes. By employing symmetry operations (mainly reflection), the method enables to generate dissimilar edge connections, giving rise to several ways for the cubic building blocks to reconfigure. While the paper draws inspiration from previous studies, such as origami-inspired transformable metamaterials and others in the literature, it stands out with its diverse range of reconfigurations and interesting demonstrations, showcasing its versatility.

The paper, however, does have some significant issues, particularly with the way the contents, including the Supplementary Information (SI), are structured, described, and explained. These issues are further exacerbated by several sentences that are flawed both grammatically and in terms of contents with several redundancies that make the text at times unfocused. Regrettably, these aspects significantly detract from the potential of the work to the point where the overall quality - as it stands - falls below the standards expected for a publication in high-impact journals.

Below are some of the issues.

Description of the challenges. The sentence below is quite convoluted and does not clearly convey the main challenges.

One of the primary challenges resides in the tradeoff between theoretically allowable shape-morphing versatility and practical controllability in terms of actuation, which could largely hinder the broad applications of shape-morphing structures in reconfigurable architecture, metamaterials, and robotics (see Supplementary Note 1.1 and Supplementary Table 1 for summary of representative reconfigurable systems).

Table 1 in Supplementary Note 1.1 is not explained. Only data are reported therein, but neither the manuscript nor the SI refers to that table 1 (pag 7 in the SI). In addition, this table of the SI can barely be read as the font size is extremely small. Also, table S1 (radar chart) is provided, but there is no text/explanation that refers to it.

The term versatility needs to be clearly defined. Although the authors state, "The versatility of shape morphing is intricately linked to a structure's mobility," it is not clear whether versatility has a broader connotation that extends beyond the degrees of freedom. Since this term is a key hallmark brought forward as an advantage over the literature, it must be defined clearly.

The term frustration is also called in the paper multiple times when the characteristics of the proposed origami are compared to those of other works. The concept is very well-defined in the literature. However, the authors seem to describe frustration-free kinematic paths as collision-free kinematic paths. This seems to be somewhat misaligned with the notion of frustration.

The term "combinatorial design capability" needs more precision. What does combinatorial capability refer to here?

The use of the term “programmed”. The authors repeatedly write that other structures cannot be programmed, but the use of this term is questionable in this context. Programming occurs at the design stage, and the design of several works cited can be tuned by design—i.e., it can be programmed. What it cannot do is be reprogrammed post-fabrication. It seems here that there is a misunderstanding of this important distinction.

Fig. 3 is a large figure encompassing multiple subfigures, each conceived and included to convey a given message. The text, however, refers to this figure only once, very briefly, without (quite surprisingly) explaining its contents. One wonders why the figure is included in the main manuscript if not much text is devoted to describing it.

Figure 4 includes several small insets with small labels /legends that are difficult to read and interpret.

Supplementary notes.

The whole text of the supplementary notes, which seems to be given to demonstrate and explain the differences and novelties of the work, is not well-structured. It seems as though the authors assign to these notes an important role, considering especially their length. However, the paragraphs are not cohesively merged and written in a structured sequence. Several parts are repeated differently, making the text lengthy and unclear. It is not straightforward to appreciate a structure of contents. In addition, there are important paragraphs, such as the one below, where a comparison is made with existing strategies, but no reference is given to them. The lack of references therein makes the comparison unfeasible. The reader is left with no means to verify the authors' claims.

Continuous structural form-based reconfigurable systems. Essentially, these design strategies and our work are two different schemes for reconfigurable and shape morphing structure designs. These methods generally aim for construct functional structures ..

SI and manuscript.

Some sentences in the supplementary notes are repeated verbatim in the introduction. I recommend avoiding this practice. The SI should complement and not repeat the text in the main manuscript.

In general, the text includes convolute explanations - uneasy to follow, and sometimes compound with English flaws, such as here

- We attribute this to the fact that the hinges have to be uniquely placed with special relative spatial positions to guarantee the level-1 structures with larger number of initial structural DOFs which will result with some structural constraints and hinder the design possibilities..
- flipping the 8-cube level-1 structures is special cases of manipulating the hinge connections at the base level.
- the first category (Category 1) structure is based on the truth that the four level-1 structures (schematically highlighted with blue color in Supplementary Fig. 5B) are positioned with same spatial postures and with both the xz-plane and yz-plane symmetries
- Different with the Category 1
- other level-2 structures can thus be classified with more categories.
- To more clearly see the shape morphing process, we refer to the graph theory applied in computer science where to build ordered data/dataset tree, and construct the corresponding data-tree like diagram applicable for all the proposed hierarchical origami metastructures in this work.
- we find that by pre-assigning a local Cartesian coordinate systems
- a comprehensive view of the continuously evolving shape-morphing configurations diagram > recursive sentence
- Most importantly, once designed, almost all previous works started from this method lack design programmability and deficient for inverse designs? A verb is missing in the second half of the sentence
- these structures exhibit with only one reconfiguration degree of freedom, exhibit with??
- These combinatorial hierarchical mechanisms is

- Comparison between selected four combinatorial categories of level-2 .. on the number

Other typos

our work outperform the > outperforms

single reconfiguration degree of freedom, which constrict > constricts

looped kinematic mechanisms has been demonstrated > have

programed >

hybrid architectures by assembly the kinematic cube >> by assembling ?? not sure the meaning here.

advantages: Firstly,

The transition kinematics describe > describes

combinatorically hinging two adjacent cubes at four pair > pairs? But what the term

"combinatorically" denotes here

The use of truth and truthfully should be carefully pondered in scientific writing. These terms can be adopted when mathematical proof is given. It doesn't seem that this is the case.

Response to reviewers' comments:

We thank both reviewers' insightful and constructive comments for improving the quality of this work. All the revisions are highlighted in **GREEN COLOR** in both main text and SI.

Reviewer #1 (Remarks to the Author):

First, this reviewer would like to apologize for the long delay in providing a review. There were some personal matters that interfered with being able to provide a timely evaluation of this paper. This paper presents a concept for hierarchical shape morphing with block-like kirigami structures. The paper is well written, the research is thoroughly performed, and the work presents novel ideas worthy of publication.

Response: We thank the reviewer for the very positive comments of our work. Following the reviewer's profound and inspiring comments, we carefully address each point with detailed responses shown as below.

First, a comment regarding the overlap with the authors' previous work presented in reference #36 – there is some overlap with the previous work in terms of concepts, methods, capabilities, and overall execution. The systems are physically similar, have a large range of shape morphing geometries and require few DOFs to reconfigure. This overlap is not ideal especially when attempting to publish this work in a premier level journal such as Nature Communications.

With that said however, this reviewer does not see this overlap as a deal-breaker, and in fact the new paper is in many ways distinct and novel compared to the previous publication. The hierarchical concept for arranging and reconfiguring the systems is newly presented here and is quite different from the ad-hoc approach presented previously. Furthermore, there is now a more systematic approach for the design of the systems through hierarchical linkages. This paper also presents novel concepts for locomotion of the block like systems. These aspects are likely sufficient to justify the work to be published, however the authors should consider the following comments which could further strengthen and distinguish the paper.

Response: We sincerely thank the reviewer for acknowledging the distinction, advancement, and novelty of this work compared to our previous work, as well as all the constructive comments for further strengthening this work.

We have thoroughly and rigorously addressed all the following solid and suggestive comments point by point.

For clarity, we have labeled all figures in the response letter as **Figure R n** ($n = 1, 2, 3, \dots$).

1 First, the paper needs to take a more balanced and measured approach, especially since this is not the first paper with the theme of block-like morphing matter. It would be helpful if the paper discusses limitations of the work, and discuss areas for future work --- This is an exciting area for morphing matter, so what's next? --- For example, the discussion of using tetrahedrons, prisms and other shapes is done exceptionally well, and other current frontiers could be discussed (perhaps without requiring a study that is in so much in depth). The next four comments, all somewhat related, could perhaps be highlighted as challenges/limitations/opportunities – some could be

explored here, or they could be discussed as topics for future study. Other discussions of limitations and future work would also be appreciated.

Response: We thank the reviewer for the excellent comment on discussing the limitation and the outlook of the morphing matter. Following reviewer's suggestions, we briefly discuss the limitations of our current work and future potential research directions of the block-like shape morphing matter. Additionally, we provide specific responses to all the following four comments.

1. Limitations of our current work

In the revised version, in Discussion section, we added one paragraph discussing the limitations of our current work as below:

“We note that there are several limitations of this work. First, the load bearing capacity of some reconfigured 3D architectures is still limited, which could hinder their practical engineering and structural applications, especially at meter scales. The load bearing capacity is dependent of not only the transformed architectures (see the free body diagrams of force analysis for example in Supplementary Fig. 23), but also the bending stiffness of both cubes and hinges and the structural designs of the hinges. The hinges are imitated with 3D printed soft rubber-like materials or tapes with low bending rigidity that facilitate the bending and rotation motion but sacrifice the load-carrying capabilities. The load bearing capacity could be improved by using stronger materials with high bending rigidity or locking hinges or devices at either 90° or 180° folded angles. Second, the shape-morphing capability for robotic applications is limited to multi-gait motion demonstrated in this work. How to leverage the rich shape-morphing capability for diverse and adaptive robotic locomotion in unstructured environments remains to be uncovered. Third, the demonstration of self-deployment and self-reconfiguration is limited to centimeter-scale prototypes while the meter-scale demo is done manually due to the limitation of both power and servomotors. At large scales, the heavier self-weight of cubes cannot be neglected, which requires high-torque servomotors and high-power batteries to generate sufficient torque output to counter the gravity and drive the folding.”

2. Discussions on future works

Based on our current findings, we envision future research directions in both design and application aspects.

2.1 Design

As the reviewer acknowledged, the generic hierarchical design in this work can be generalized to structural elements with versatile 2D and 3D versatile shapes originally discussed in the Discussion part of the manuscript, which will not be repeated here.

2.1 Involving mechanics into our proposed systems

Our current work considers only the kinematics. As remarked by the reviewer in Comment #4, involving mechanics by taking into account the physics of the model can provide more insights into the mechanical properties, performance, and applications of our proposed systems. Therefore, we believe that the following several directions can be explored in future studies.

2.1.1 Deformation behaviors of the hierarchical origami structures with elastic hinges. In this direction, the hinges will be assumed with nontrivial stiffness and can be largely stretchable, bendable, and twistable. Then, the reconfiguration features of our proposed hierarchical origami metastructures can be revisited. Enabled by the complex deformability of the hinges, novel deformation features can be thoroughly explored by building related mechanics models. We envision that our systems can maintain their original reconfiguration features while achieving some new configurations due to the hinges deformability, for example reconfigured shapes with multiple structural stability can be achieved in higher level structures.

2.1.2 Using as structural units to build 3D mechanical metamaterials/metastructures. From Supplementary Figs. 6, 8, 12, and 13, we observe that both level-2 and level-3 structures can reconfigure into configurations that deform into divergent configurations through bifurcated path branches. These configurations possess ideal boundary conditions for tessellation into periodic 3D metamaterials/metastructures. Therefore, we envision another future direction focused on searching for optimal modular hierarchical origami metastructures whose reconfigured configurations are suitable for building 3D metamaterials/metastructures with bifurcation deformation features. To the best of our knowledge, there have been limited research efforts using bifurcated structural units to create 3D mechanical metamaterials. We believe this field shows promise and is worthy of further study. Additionally, by replacing the pure rotating hinges and rigid cubes with deformable ones (e.g., stretchable, bendable, twistable), this novel type of mechanical metamaterial may not only benefit from the bifurcated structural units but also exhibit more unconventional mechanical properties.

2.1.3 Creating kinematic bifurcation enabled multistable mechanical metastructures. In our current work, kinematic bifurcation enhances the reconfiguration capability of our systems through more kinematic paths. Intrinsically, this feature allows us to combine the physics of the system, such as boundary conditions, hinge stiffness, and cube deformability, to create novel planar mechanical metastructures with multistability. These structures are promising with potential applications for mechanical computing and information processing.

2.1.4 Extending to conventional origami structures. Since rigidly foldable origami can be analyzed as kinematic mechanisms, we envision that our hierarchical design scheme can fundamentally broaden the design space of the origami family. For example, beyond the thick panel presented in our work, future research could focus on extending the structural hierarchy concept to previously designed conventional 2D origami structures, such as those with Miura-ori, Kresling, and curve-creased fold patterns. Further exploration could lead to creating novel origami-based mechanical metamaterials and/or metastructures with new mechanical properties and shape-morphing features.

2.2 Applications

2.2.1 Reconfigurable modular aerospace structures and robots. In Figs. 6 and 7, we have demonstrated the capability of our proposed hierarchical origami metastructures in designing autonomously reconfigurable robots and architectures. Based on these results, we envision that our structures hold promise for constructing reconfigurable aerospace structures and robots that can change into on-demand shapes to adapt to fast changing application scenarios, and also avoid damage from harsh situations. However, this research concept is still in its infancy. Future studies

in this direction can focus on (1) robust structural design, (2) reliable control system construction, and (3) embodied intelligence for practical uses.

2.2.2 Reconfigurable civil and architectural structures. In Fig. 8a, we have discussed the potential applications of our proposed hierarchical origami metastructures as civil and architectural shelter structures. However, we note that the reconfigured shapes for shelter structure use in Fig. 8a are not ideal practically, especially given their smaller enclosed internal volumes and bulky structural frames. Therefore, future studies in this direction can focus on (1) searching for suitable structural designs with optimal load-bearing capability and efficient enclosed internal volume, (2) rational analysis of the structural stiffness of the searched structures, and (3) practical verifications by fabricating physical prototypes with on-site mechanical tests. When searching for optimal structures, we suggest two strategies: (a) using different shaped structural elements (for example, replacing the cube with thick plate), and (b) trying different structural motifs (for example, the level-3 structure with <4R, 4R, 4R> type of structural motif).

Along with the abovementioned research areas, we believe new fundamental theoretical modelling framework may also be required and built accordingly in future studies.

Based on above discussions about the limitations and potential future research areas, we now can take a balanced and measured approach of our work. Correspondingly, we add some discussions in the “Discussion” part of the main text regarding the future outlook.

Revisions: We added

“ Moving forward, these limitations also open new opportunities for future researches in morphing matter. First, this work explores only a small region of the tremendous design space in morphing matter to showcase its potential. The vast combinatorial folding patterns arise from the combinatorial connections in the base units, as well as within and across each hierarchical mechanism (Supplementary Fig. 3). These combinatorial hierarchical mechanisms are generalizable and can be applied to construct similar reconfigurable hierarchical metastructures composed of any shape-morphing spatial closed-loop mechanism for easy actuation and control yet rich shape morphing. For example, the cube units can be replaced by other composed geometrical shapes, such as thick plates with substantially reduced thickness dimension, tetrahedrons, and triangular-shaped prisms, or extended to genuine volumetric 3D structures (examples are provided in Supplementary Figs. 27-28, with more details in Supplementary Note 12).

Second, this work focuses on exploring the reconfiguration kinematics of the hierarchical origami systems by modeling the system as idealized hierarchical rigid mechanisms and neglecting the deformation in both the cubes and hinges. However, in scenarios when such elastic deformation are non-negligible, similar to the non-rigidly deformable origami metamaterials in origami engineering, the over-constrained looped kinematic mechanisms become energy scale dependent, considering the potentially involved complex deformation in the cubes, hinges, and architectures during reconfiguration such as bending, stretching, twisting, and shearing or combined. Consequently, it will transform the rigid mechanisms into both reconfigurable and deformable architected materials and structures, which couples kinematics with mechanics. Such coupling will enrich new kinematics, mechanics, transformed configurations, reconfiguration paths, and

reprogrammable mechanical behaviors such as multistability and stiffness anisotropy. Specially, how the energy scale affects the kinematic bifurcated paths and how the coupled kinematic bifurcation and elasticity change both the reconfigurations and mechanical responses of bifurcated mechanical metamaterials remain to be uncovered. We envision such studies could also find broad applications in reprogrammable mechanical computing, mechanical memory, and mechanical metamaterials.

Third, considering these multi-capabilities in conjunction with scalability, modularity, and disassemblability, we envision diverse applications in robotics, architecture, and even in space. **Fig. 7b** conceptually illustrates potential applications in multitask adaptive shape-morphing space robots and habitat (Supplementary Movie 8). The hierarchical origami architectures could be deployed with largely increased exposed surface areas for enhanced solar energy harvesting, and reconfigured to avoid debris collision or accommodate more docking stations. It could also serve as reconfigurable space habitat or be des-assembled into modular robots for multitask exploration. For large-sized structures, the feasibility of actuation in a space environment is considerably higher, primarily due to the absence of gravity and the absence of ground-based collisions that can impede complex shape-morphing processes on Earth. ”

2 The paper uses a concept of what are technically over-constrained linkages. In practice, if such linkages are fabricated with high precision and with rigid links and hinges, then they become highly susceptible to locking and not following their kinematics properly (e.g. due to small imperfections). In contrast most such system are instead fabricated with some ‘slack’ or elastic deformation built in which allows the systems to move more freely. The issue with this slack is that it can lead to rather floppy systems with little stiffness. From the videos it appears that the systems here are indeed fabricated with some slack, and are somewhat floppy. Can the authors discuss this slack in more detail? Are there issues with the systems following their kinematic paths? Are the systems sometimes overly floppy leading to unwanted deformations? Is there some systematic guidance that they can give related to this?

Response: We thank the reviewer for the insightful comments.

We agree that some slack might arise due to imperfect fabrications, potentially leading to the structure exhibiting floppy deformation modes. These slacks could be eliminated by constructing the structure with high-precision rigid links and hinges. As pointed out by the reviewer that structures without slack may sometimes face hinge locking issues, especially when actively actuated. We elaborate on these points explicitly below.

As illustrated in Fig. 4a and Supplementary Fig. 7, we manufactured physical prototypes of <4R, 8R> type level-2 structures using a multi-material 3D printing method for concept verification. Cube elements and hinges were printed separately with rigid and soft rubber-like materials. To ensure almost free rotations, the soft hinges needed to be fabricated with a certain length (1mm) and relatively small thickness (0.6mm for "little stiffness"). Consequently, except for pure rotations, these practically fabricated hinges might experience some twisting deformations, potentially causing the structure to deviate from their ideal kinematic paths. However, these path deviations occur only in complex reconfiguration processes (for example from Configuration **M**₇ to Configuration **M**₁₃ of the level-2 structure demonstrated in Fig. 4a, now updated as Fig. 3a), and

are suppressed when the reconfiguring structure exhibits structural symmetries (for example from Configuration \mathbf{M}_D to Configuration \mathbf{M}_E in Fig. 4a always with x-y and z-y plane symmetries).

Furthermore, hinges with small twisting deformations do not significantly deviate the system from the original reconfiguration paths (or become overly floppy), ultimately leading to a configuration only slightly different from the ideal one. New reconfiguration kinematics may occur in the system and can be rationally determined if hinges are not stretchable. For instance, the 8R level-1 structure shown in Fig. 5b and Fig. 5c can maintain as a looped mechanism even the hinges are also twisted. The system's reconfiguration kinematic can still be predicted based on the Denavit-Hartenberg theorem by equation

$$\prod_{m=1}^8 T_m = I$$

wherein T_m ($m = 1, 2, \dots, 8$) as transformation matrix only with variable of hinges rotation angle γ_m ($\gamma_m = (\gamma_{mx}, \gamma_{my}, \gamma_{mz})$, see Figure R1).

Specially, when no or negligibly slight slacks occur in hinges, there will be only one nonzero component in γ_m . Thus, to determine the system's reconfiguration kinematics is rather easy and straightforward. However, when the slack is non-negligible, all three nonzero components in γ_m become nonzero, significantly improving the system's reconfiguration degrees of freedom and making the resolving process a complex nonlinear problem. In this situation, more advanced computational methods need to be explored. This intriguing problem remains open for future research. However, since in real applications hinge's rotation is mainly induced by its bending deformation, the slack can therefore be avoided by fabricating hinges with a low ratio of bending stiffness to twisting stiffness.

b Link #k of level-1 structure

c Local coordinate systems

Figure R1 The reconfigured 8R level-1 structure in Fig. 5b and Fig. 5c.

For systems fabricated with high-precision rigid links and hinges, slack can be eliminated entirely, and the system can rigorously follow its reconfiguration kinematic path. This was experimentally demonstrated by us through manufacturing both level-1 and level-2 structures with 3D printed rigid cubes and rigidly rotatable hinges (metal rods as passive hinges and electrical servomotors as active hinges; see Fig. 6a, now updated as Fig. 5a). According to our experimental results displayed in Figs. 6 and 7, the systems can smoothly reconfigure into all desired configurations without any locking issues.

Regarding the reviewer's concern about system locking and not following the reconfiguration path properly, we believe this issue can be feasibly solved by using actuating servomotors with large enough torque output to overcome the resistance induced by any experimental imperfections.

Revision: In the Discussion Section, we added a new paragraph discussing the slack as below:

“Our design strategy combines structural hierarchy with over-constrained looped kinematic mechanism without considering elastic deformation in the hinges and cubes. Practically, the elastic deformation or slack, especially in the hinges, could cause the system to be floppy or potentially deviate from the desired non-bifurcated and/or bifurcated kinematic paths. As demonstrated in the multimaterial 3D printed level-2 structure in Fig. 3a, the soft hinges are printed thin with little stiffness to ensure almost free rotation. Thus, in addition to bending for rotation motion, the hinges also undergo certain twisting deformation, potentially causing the structure to deviate from their ideal kinematic paths. However, deviations occur only during the complex reconfiguration processes, e.g., from configuration M_7 to M_{13} in Fig. 3a. Such deviations are suppressed when the reconfiguring structure exhibits structural symmetries, e.g., from configuration M_D to Configuration M_E in Fig. 3a preserving x - y and z - y plane symmetries. The slack can be avoided by fabricating hinges with a low ratio of bending stiffness to twisting stiffness. This will help to suppress its twisting deformation to follow the kinematic paths without making the structure overly floppy. For systems fabricated with high-precision rigid links and hinges, slack or elastic deformation can be minimized or eliminated, as demonstrated in the prototype of both level-1 and level-2 structures with 3D printed rigid cubes and rigidly rotatable hinges in Fig. 5a. Similar to studied 2D rigidly foldable origami structures, the reconfiguration kinematics of the system becomes energy scale independent. Thus, the system can rigorously follow its bifurcated reconfiguration kinematic path via fewer number of actuation hinges to smoothly reconfigure into all desired configurations without any locking issues as demonstrated in Figs. 5-6.”

3 Highly related to #2 above, the systems here, much like origami/kirigami rely on bifurcations to go into the desired kinematic paths. This can be especially challenging and tricky with complex systems. For example ‘Self-folding origami at any energy scale’ by Pinson et al. in Nature Communication, and other papers in the field have discussed these issues in some detail. Can the practicality of entering a desired path be briefly explored here? Are the systems always reliable, or are there issues with entering some of the potentially higher energy modes? A brief discussion related to this issue would help.

Response: These comments are very inspiring and helpful.

We total agree with the reviewer that entering the desired kinematic paths for complex elastic origami systems are difficult, especially when these origami systems are not completely rigidly foldable and with deformable face panels. However, even with kinematic bifurcations we think it is not an issue for our designed systems. We explain this in detail below. Note that since our proposed structure reconfigure only relied on hinge rotation, severe floppy mode will not be considered here.

In our designed hierarchical origami metastructures, the cube shaped links are always assumed and fabricated to be totally rigid without any deformations. Therefore, similar to previous 2D rigidly foldable origami structures with non-deformable face panels, the reconfiguration kinematics of our proposed structures are energy scale independent. Regardless of whether the hinges are fabricated to be deformable with small or finite energies, the reconfiguration kinematics of our systems can always be analytically or numerically determined based on the Denavit-Hartbenberg theorem, (i.e., the equation $\prod_m^k T_m = I$ with k related to the type of structural hierarchy motif illustrated as Supplementary Fig. 3).

In Supplementary Notes 4-7, we have demonstrated that the reconfiguration kinematics of all reconfiguration paths, including the bifurcated ones, of our systems are reflected in the particular relations of some or all hinge folding angles. Therefore, entering a desired kinematic path means actuating a certain number of hinges together in practice. Moreover, leveraging the looped structural features, we've shown that the reconfiguration degrees of freedom of our proposed level-1 and level-2 structures are fewer than 3. Consequently, in real applications, we only need to actively actuate no more than three hinges (which can be different sets) to access all desired reconfiguration paths.

These conclusions can be verified by the feasible reconfigurations of our selected level-1 and level-2 structures fabricated with elastic or totally freely rotated hinges. Particularly, as shown in Figs. 6b-6e and Supplementary Fig. 7b, we can accurately reconfigure the 8R level-1 structures into different paths by selectively rotating 2, 6, or 8 hinges together, see also the **Figure R3** below. When the level-1 structure reconfigures into the bifurcated configuration state, for example, the configuration with all eight hinges' rotation angles equal to 90° (i.e., bifurcation point 2 illustrated in Supplementary Fig. 7b), it can further enter the three bifurcated paths by monotonically rotating two different sets of four hinges (i.e., bifurcated paths ② and ③ in the figure below) while maintaining the other four at 90° , or continue rotating all eight hinges (path ①).

b Reconfigurations of level-1 system in **Figure 4a**

Figure R2 Experimental verification for all the reconfiguration paths of the level-1 structure in Supplementary Fig. 7b.

Beyond the level-1 structure, the level-2 structure can similarly execute all its reconfiguration paths by selectively controlling part or all of the level-1 and level-2 hinges. Representative reconfiguration details are presented in Fig. 5 and Supplementary Fig. 18.

Hence, we can confidently conclude that entering the desired kinematic paths, including the bifurcated ones, is not an issue for our systems, regardless of energy modes.

Revision: In the Discussion section, we briefly discussed the energy-related bifurcation and paths as below in both discussions and outlook for future work:

“For systems fabricated with high-precision rigid links and hinges, slack or elastic deformation can be minimized or eliminated, as demonstrated in the prototype of both level-1 and level-2 structures with 3D printed rigid cubes and rigidly rotatable hinges in Fig. 5a. Similar to studied 2D rigidly foldable origami structures, the reconfiguration kinematics of the system becomes energy scale independent. Thus, the system can rigorously follow its bifurcated reconfiguration kinematic path via fewer number of actuation hinges to smoothly reconfigure into all desired configurations without any locking issues as demonstrated in Figs. 5-6.”

“ Second, this work focuses on exploring the reconfiguration kinematics of the hierarchical origami systems by modeling the system as idealized hierarchical rigid mechanisms and neglecting the deformation in both the cubes and hinges. However, in scenarios when such elastic deformation are non-negligible, similar to the non-rigidly deformable origami metamaterials in origami engineering, the over-constrained looped kinematic mechanisms become energy scale dependent, considering the potentially involved complex deformation in the cubes, hinges, and architectures during reconfiguration such as bending, stretching, twisting, and shearing or combined. Consequently, it will transform the rigid mechanisms into both reconfigurable and deformable architected materials and structures, which couples kinematics with mechanics. Such coupling will enrich new kinematics, mechanics, transformed configurations, reconfiguration paths, and reprogrammable mechanical behaviors such as multistability and stiffness anisotropy. Specially, how the energy scale affects the kinematic bifurcated paths and how the coupled kinematic bifurcation and elasticity change both the reconfigurations and mechanical responses of bifurcated mechanical metamaterials remain to be uncovered. We envision such studies could also find broad applications in reprogrammable mechanical computing, mechanical memory, and mechanical metamaterials. ”

4 Again, related to the 2&3; but with a slightly different direction: The work here is focused purely on the kinematics of the systems. This focus is fine for now; however, the fields of origami/kirigami has moved to performing much more advanced mechanics based simulations. These simulations take into account more of the physics of the model including stiffness and deformations, and are well suited to dealing with some of the challenges in 2&3 above, and more. Furthermore, mechanics based models can give substantially more insight on physical behaviors, performance, and applications. Using such models to study these systems is a natural next step related to this work. Again, maybe this is something that does not need to be done here, but could be discussed as future work.

Response: We thank the reviewer for the insightful comments.

We totally agree with the reviewer's perspective on the significance of introducing mechanics into our proposed hierarchical origami structures. This integration promises to furnish us with deeper insights into their physical behaviors, performance characteristics, and potential applications. We sincerely appreciate the reviewer's valuable suggestions in this regard.

Indeed, it aligns closely with one of our planned future endeavors to infuse mechanics into our systems to unearth novel findings. For instance, we are presently engaged in exploring how the stiffness of hinges and the imposition of boundary constraints influence the deformation characteristics of our proposed structures. Furthermore, leveraging unique hierarchical origami structures as building blocks, we are actively designing periodic structures endowed with multistable deformation features, with the intent to explore their applications in mechanical computing and information processing.

Revision: Following the reviewer's insightful suggestion, we have included this direction as one of our future works. Further details can be found in our detailed responses to Comment #2 and 3, specifically under "2.1.2 Involving mechanics into our proposed systems."

5 The stiffness and load bearing capability of the systems here is not quite that impressive compared to other origami/kirigami systems. The authors should distinctly highlight that as a current limitation. Perhaps more detailed modeling as in (4) together with the vast array of possible geometries can offer some unique geometries that come closer to other origami/kirigami designs. It is likely alright if the topics in 2-5 above are not fully explored/addressed, however they should be discussed at least in some detail in this paper.

Response: We thank the reviewer for the solid comment.

Following the reviewer's suggestion, we have incorporated the limitation of load-bearing capability as one of the current constraints in our work.

Revision: In the Discussion section on limitations, we added

“We note that there are several limitations of this work. First, the load bearing capacity of some reconfigured 3D architectures is still limited, which could hinder their practical engineering and structural applications, especially at meter scales. The load bearing capacity is dependent of not only the transformed architectures (see the free body diagrams of force analysis for example in Supplementary Fig. 23), but also the bending stiffness of both cubes and hinges and the structural designs of the hinges. The hinges are imitated with 3D printed soft rubber-like materials or tapes with low bending rigidity that facilitate the bending and rotation motion but sacrifice the load-carrying capabilities. The load bearing capacity could be improved by using stronger materials with high bending rigidity or locking hinges or devices at either 90° or 180° folded angles.”

Addressing the reviewer's comments 2-5, we have made concerted efforts to provide detailed responses, as elucidated above. Furthermore, we have succinctly summarized these responses by integrating additional discussions into the "Discussion" section of the main text.

Reviewer #2 (Remarks to the Author):

The paper presents a method that uses the combinatorial pairing of edges between adjacent cubes to construct rigid-foldable origami that can morph into multiple reconfigurations. The strategy (denoted as hierarchical) explores the permutations that exist for connecting edges of neighboring cubes. By employing symmetry operations (mainly reflection), the method enables to generate dissimilar edge connections, giving rise to several ways for the cubic building blocks to reconfigure.

While the paper draws inspiration from previous studies, such as origami-inspired transformable metamaterials and others in the literature, it stands out with its diverse range of reconfigurations and interesting demonstrations, showcasing its versatility.

Response: We thank the reviewer for appreciating our work. All the revisions are highlighted in GREEN COLOR in both main text and SI.

The paper, however, does have some significant issues, particularly with the way the contents, including the Supplementary Information (SI), are structured, described, and explained. These issues are further exacerbated by several sentences that are flawed both grammatically and in terms of contents with several redundancies that make the text at times unfocused. Regrettably, these aspects significantly detract from the potential of the work to the point where the overall quality - as it stands - falls below the standards expected for a publication in high-impact journals.

Below are some of the issues.

Response: We thank the reviewer for the suggestive comment. We apologize for any lack of clarity or organization in our manuscript and Supplementary Information. Our intention is always to provide clear and accurate information.

Following the reviewer's comments, we have diligently enhanced our manuscript by providing more comprehensive explanations and descriptions in both the main text and Supplementary Information. We have meticulously refined the entire manuscript to ensure conciseness, removing any redundant content and correcting all grammatical errors.

Description of the challenges. The sentence below is quite convoluted and does not clearly convey the main challenges.

One of the primary challenges resides in the tradeoff between theoretically allowable shape-morphing versatility and practical controllability in terms of actuation, which could largely hinder the broad applications of shape-morphing structures in reconfigurable architecture, metamaterials, and robotics (see Supplementary Note 1.1 and Supplementary Table 1 for summary of representative reconfigurable systems).

Response: We sincerely thank the reviewer for pointing this out and helping us improve our work. We are sorry that we did not convey the challenges of shape morphing structures clearly. To avoid any further confusions, we rewrite this sentence by adding more explanations in both main text and supplementary information.

In main text, we modify this sentence as

“One of the primary challenges resides in the tradeoff between theoretically allowable shape-morphing versatility, which encompasses the quantity and diversity/type of reconfigured shapes, and practical controllability in terms of actuation. For instance, while previously reported structures^{11,23,24,26-28,36} have demonstrated the ability to change into a vast number of distinct shapes, they often require exceedingly complex actuation and control systems. This complexity can render the shape morphing process tedious, time-consuming, and energy-inefficient. On the other hand, certain structures may exhibit simpler reconfiguration kinematics^{3,5-7,15,31,41,47,48}, enabling them to feasibly attain desired shapes. However, their unique structural forms may significantly limit the achievable reconfigured shapes within few specific categories. These challenges, along with others such as complex reconfiguration kinematics, poor re-programmability, lack of inverse design capability, and limited functionality of the reconfigured shapes, as summarized in Supplementary Table S1, could significantly impede the broad applications of shape-morphing structures in areas such as reconfigurable architecture, metamaterials, and robotics (see more details in Supplementary Note 1).”

And in the supplementary information, we update the supplementary text on Page 2 and Page 3 from Line 35 to Line 70 as

“Tradeoff between shape morphing versatility and controllable actuation: Versatile shape-morphing capability is crucial for enabling multifunctional applications in engineered structures. This capability is defined by the availability of a vast number and types of achievable reconfigured shapes suitable for various applications. Consequently, structures with versatile shape-morphing capability can promptly adapt to fast changing external environments and diverse application scenarios by reconfiguring into on-demand shapes.

The versatility of shape morphing is directly linked to a structure’s mobility, as indicated by its degrees of freedom (DOF). Theoretically, structures with a higher number of DOF tend to offer more extensive shape morphing capabilities, allowing them to assume numerous configurations. However, higher DOF can pose practical challenges for actuation. In real-world applications, each DOF requires a corresponding distributed actuator. Excessive DOF can lead to increased hardware requirements and complicate the associated control systems (software). Consequently, this complexity hinders the broad applicability of such structures. This tradeoff between theoretically feasible shape morphing versatility and practical controllability in terms of actuation remains a significant challenge, particularly in fields like reconfigurable metastructures and robotics.

To address this challenge, numerous studies have been dedicated to overcoming the tradeoff between shape morphing versatility and controllability. However, as summarized in Supplementary Table S1 and discussed in the following section, previously designed shape morphing structures have typically exhibited one of two limitations: they either sacrifice shape morphing versatility to achieve feasibility in control systems and actuation, or they integrate too many actuation systems, resulting in inefficient configuration changes and prolonged reconfiguration times. Thus far, solutions have prioritized either versatility or actuation, but achieving both remains elusive. Specifically, the realization of versatile shape morphing structures with a limited number of easily controllable actuation degrees of freedom (DOFs) has yet to be achieved. This limitation significantly impedes their applications in areas such as reconfigurable metamaterials, robotics, building construction, and space infrastructure, where rapid, energy-

efficient shape changes with simple kinematics and adaptable configurations are essential for fast changing application scenarios.

By incorporating closed-loop mechanisms within and across each hierarchical level to minimize the number of actuated DOFs, our proposed hierarchical origami metastructures can eliminate the need to actuate each individual DOF as in previous reconfigurable origami metamaterials and robots but exhibiting promising shape morphing versatility with fewer reconfiguration DOF and simple actuations. As a result, this advancement opens up possibilities for widespread applications, including the construction of highly efficient reconfigurable robots, as well as the rapid deployment of reconfigurable architecture on Earth and potentially in space, serving as multitask space robots and habitats.”

Table 1 in Supplementary Note 1.1 is not explained. Only data are reported therein, but neither the manuscript nor the SI refers to that table 1 (page 7 in the SI). In addition, this table of the SI can barely be read as the font size is extremely small. Also, table S1 (radar chart) is provided, but there is no text/explanation that refers to it.

Response: We thank the reviewer for the constructive comment.

To explicitly report the Supplementary Table S1, we add more information to explain and appropriately refer it in both main text and supplementary information.

In main text, we refer to the Supplementary Table S1 on Page 3 from Line 53 to Line 58 in the updated sentence as

“These challenges, along with others such as complex reconfiguration kinematics, poor reprogrammability, lack of inverse design capability, and limited functionality of the reconfigured shapes, as summarized in Supplementary Table S1, could significantly impede the broad applications of shape-morphing structures in areas such as reconfigurable architecture, metamaterials, and robotics (see more details in Supplementary Note 1).”

, and on Page 4 from Line 80 to Line 84 of the sentence

“Although these modular origami and robotic structures offer enhanced shape-morphing capabilities, they typically require control and actuation systems for each module. This complexity results in lengthy and intricate reconfiguration steps, as well as complex and time-consuming actuation, morphing kinematics, and reconfiguration paths, primarily due to their redundant DOFs^{11,25-27,35,36} (Supplementary Table S1 and S1 and related discussions in Supplementary Note 1).”

In supplementary information on Page 3 to Page 8 from Line 81 to Line 226, we rewrite the Supplementary Note 1.1 with contents as

“Supplementary Note 1.1 Comparison with the state-of-the-art reconfigurable systems
In Supplementary Table S1, we provide detailed comparisons between our work and previous shape-morphing structures. We have selected the five most representative types of shape-morphing structures, which include

Moreover, the font size of the Supplementary Table S1 is enlarged for readability, and a detailed caption is added for explanation. We update the Supplementary Table S1 along with the radar chart as

References	Reconfiguration DOF	Reconfiguration time	Reconfiguration steps	Kinematics	Achievable shapes	Re-programmability	Inverse Design	Functionalities	
Shape morphing structures with continuum structural forms	3	Infinity	106s	1	Nonlinear bending	3D curved surfaces	✓	✓	Shape morphing
	5	Infinity	90s	1	Nonlinear bending	3D curved surfaces	✓	✓	Shape morphing
	6	Infinity	30-130s	1	Nonlinear bending	3D curved surfaces	✓	✓	Shape morphing, robotic running
	35	Infinity	<1s	1	Nonlinear bending	3D curved surfaces	✓	✓	Shape morphing, object manipulation
Kinematic Mechanism based structures	7	1	No actuation	1	Nonlinear rigid rotation	1	×	×	Shape morphing
	8	1	No actuation	1	Linear rigid rotation	1	×	×	Shape morphing
	10	1	No actuation	1	Nonlinear rigid rotation	1	×	×	Shape morphing
	11	2^N-1	No actuation	2^N-1	Linear rigid rotation	Exponentially related to unit number	✓	×	Shape morphing
	13	1	<3s	1	Nonlinear rigid rotation	4	×	×	Shape morphing
	32	1	No actuation	1	Nonlinear rigid rotation	1	×	×	Shape morphing
	42	1	No actuation	1	Linear rigid rotation	1	×	×	Shape morphing
Origami/Kirigami based structures	15	1	No actuation	1	Nonlinear rigid rotation	1	×	×	Architectural structure
	16	1	No actuation	1	Linear rigid rotation	1	×	×	Shape morphing
	18	1	No actuation	1	Nonlinear rigid rotation	1	×	×	Shape morphing
	22	1	No actuation	1	Nonlinear rigid rotation	1	×	×	Shape morphing
	26	$\gg 1$	Determined by the number of modules (10^1-10^2 s)	Nonlinearly related with number of modules	Linear rigid rotation with complex planning algorithm	Exponentially related to unit number	✓ (By changing internal connectivity)	×	Shape morphing and object transporting
	29	1	10s	1	Nonlinear rigid rotation	1	×	×	Civil shelter
	31	1	1s	1	Linear rigid rotation	1	×	×	Surgical tool
	36	1	No actuation	Exponentially related to unit number	Linear rigid rotation	Exponentially related to unit number	✓ (By changing internal connectivity)	×	Shape morphing for architectural uses
	38	1 to 7	<10s	1	Linear rigid rotation	3	×	×	Shape morphing for architected materials
	39	1	No actuation	1	Linear rigid rotation	< 10	×	×	Shape morphing for architected materials
	40	1	<10s	1	Nonlinear rigid rotation	2	×	×	Shape morphing for driving wheel
41	1	40s	1	Linear rigid rotation	1	×	×	Shape morphing for cargo frame	
48	1	<10s	1	Linear rigid rotation	<10	✓ (By changing internal connectivity)	×	shape morphing for functional exoskeletons	
Metamaterial and metastructure based	38	1 to 7	No actuation	1 to 7	Linear rigid rotation	1 to 7	×	✓	metamaterial
	39	1	1 to 10s	1	Linear rigid rotation	1	×	×	metamaterial
	47	N-1	No actuation	Linearly related to achieved shapes	Linear rigid rotation	1	×	×	metamaterial
	48	1	No actuation	1	Nonlinear buckling	1	×	×	metamaterial
Shape morphing structures assembled by discrete modules	23	N	Determined by the number of modules (10^2 s)	Exponentially related to number of modules	Repeating linear rigid rotation of all modules	3D volumetric structural profiles with infinity number	✓ (By changing internal connectivity)	✓	Shape morphing
	24	N	Determined by the number of modules (10^2 s)	Exponentially related to number of modules	Repeating linear rigid rotation of all modules	Limited to certain 3D shapes (< 10^2)	✓ (By changing internal connectivity)	×	Shaper morphing for locomotion robot and architectural structures
	27	N	Determined by the number of modules (10^1 s)	Nonlinearly related to number of modules	Repeating linear rigid rotation of all modules	Limited to linkage structures (<10)	✓ (By changing internal connectivity)	×	Shape morphing
	28	N	Determined by the number of modules (10^2 s)	Nonlinearly related to number of modules	Repeating linear rigid rotation of all modules	Limited only to 2D shapes	✓ (By changing internal connectivity)	✓	Swarming based locomotion
This work	1 to 3	< 10^2 s	< 10^1	Linear rigid rotation of low number of internal folds	Infinity due to the combinatorial design and the design generality	✓	✓	Shape morphing for architectural structures, locomotion, object transport, shelter; Metamaterials; Robotic structures	

Note 1: N is the number of structural components;

Note 2: Refs. 23, 24, 26, 27, 28, 36 and works summarized in [25] for fair comparison given their most related similarity with our work

Contd.

Supplementary Table S1 Comparison between this work and the selected five most representative state-of-the-art reconfigurable systems. Radar chart: comparison of the reconfiguration performances between our proposed hierarchical origami structures with part of previous systems (mainly those assembled by discrete modules, some origami and kinematic mechanisms based structures).

The term versatility needs to be clearly defined. Although the authors state, “The versatility of shape morphing is intricately linked to a structure’s mobility,” it is not clear whether versatility has a broader connotation that extends beyond the degrees of freedom. Since this term is a key hallmark brought forward as an advantage over the literature, it must be defined clearly.

Response: We thank the reviewer for bringing the versatility for clarification.

Indeed, versatility has a broader meaning that encompasses, but extends beyond, a structure's mobility and degrees of freedom. In this context, the versatility of shape morphing refers to a structure's capability to transform into a vast number of shapes, considering both quantity and diversity (i.e., the types/categories of the achieved geometries).

It should be noted that a structure with promising versatility in shape morphing should possess relatively high mobility or degrees of freedom to ensure it can generate a vast number of shapes. However, a structure with high structural mobility might reconfigure into numerous shapes, but these shapes may be limited to certain geometric types. For example, chain-like kinematic mechanisms (Ref. 11) and some mechanical metamaterials (Ref. 38) based shape-morphing structures exhibit high structural mobility, yet the geometry of their achieved shapes is often constrained to several types.

Following the reviewer’s suggestion, we add definitions for the versatility in main text on page 2 of the sentence from Line 44 and Line 46 as

“One of the primary challenges resides in the tradeoff between theoretically allowable versatility of shape-morphing, which encompasses the quantity and diversity/type of reconfigured shapes, and practical controllability in terms of actuation.”

The term frustration is also called in the paper multiple times when the characteristics of the proposed origami are compared to those of other works. The concept is very well-defined in the literature. However, the authors seem to describe frustration-free kinematic paths as collision-free kinematic paths. This seems to be somewhat misaligned with the notion of frustration.

Response: We thank the reviewer for the comment. To avoid any further confusions, we deleted the term of “frustration-free”.

The term “combinatorial design capability” needs more precision. What does combinatorial capability refer to here?

Response: Thanks for the suggestive comment.

For our proposed hierarchical origami structures, the cube structural elements and the entire level- n structures can be treated as rigid links. Line hinges are used to connect these rigid links together, with the number of hinges assumed to be equal to the number of links.

Given the structural topology of the cube-shaped elements and the rectangular block-shaped level- n structures, there are always multiple side pairs (e.g., the four cube side pairs of the level-1 structure and the representative side pair of the $\langle 8R, 4R \rangle$ level-2 structure shown in Fig. 2d, see details in the **Figure R4** below). This allows us to select any available side pair between two adjacent cubes or level- n structures to place a single line hinge combinatorially. For example, a structure with k cubes theoretically allows for 4^k combinatorial sets of connections, providing an extensive design space for level-1 structures (Supplementary Fig. 4).

Additionally, considering the asymmetric patterned hinges on the top and bottom surfaces across the thickness (see Fig. 2a, ii, defined along the z direction), the combinatorial design strategy can be expanded through higher-level ($n \geq 2$) structures by flipping sublevel structures around the y -axis to new postures. Based on the $\langle 8R, 4R \rangle$ type of level-2 structures, we present some explicit demonstrations in both Fig. 2f and Supplementary Fig. 5.

Figure R4 Schematics of the combinatorial design capability of our proposed hierarchical origami metastructures by rearranging hinge positions and sublevel structures spatial postures illustrated as Fig. 2d.

Thirdly, the existence of more than one structural motifs, including the 4R, 6R, and 8R looped kinematic mechanisms, allow for a third type of combinatorial design by assigning different structural motifs to each structural level. For example, if we select the 4R looped mechanism as the level-2 structural motif, we can combinatorially design three different types of level-2 structures: $\langle 4R, 4R \rangle$, $\langle 6R, 4R \rangle$, and $\langle 8R, 4R \rangle$. We have explicitly demonstrated this combinatorial design strategy in Supplementary Fig. 3.

It is important to note that all the aforementioned combinatorial design approaches can be performed simultaneously when constructing new hierarchical origami metastructures.

In fact, we already give some explanations in main text to express the combinatorial design capability of our proposed hierarchical origami metastructures, for example the contents in main text on Page 7 from Line 169 to Line 171, and those on Page 9 from Line 210 to Line 214.

To precisely express the term of combinatorial design capability, we update the main text on Page 7 and Page 9 particularly stress out its definition, i.e.,

“For two adjacent cube faces, four potential edge locations exist to accommodate hinge joints (**Fig. 2d, i**). Consequently, a structure with n cubes theoretically allows for 4^n combinatorial sets of connections, offering an extensive design space for level-1 structures (Supplementary Fig. 4). Specially, we define this multiple design possibility by the placement of hinge joints in all the leveled structures as their combinatorial design capability. As illustrated later, the combinatorial design capability of our proposed systems can be significantly expanded given the structural asymmetries and the multiple choices of structural motifs”

The use of the term “programmed”. The authors repeatedly write that other structures cannot be programmed, but the use of this term is questionable in this context. Programming occurs at the design stage, and the design of several works cited can be tuned by design—i.e., it can be programmed. What it cannot do is be reprogrammed post-fabrication. It seems here that there is a misunderstanding of this important distinction.

Response: We are sorry for the misleading. The reviewer is correct. What previous works cannot do is re-programmability post fabrication. To avoid further confusions, we update the term of “program” to “reprogram” through the whole body of the manuscript.

Fig. 3 is a large figure encompassing multiple subfigures, each conceived and included to convey a given message. The text, however, refers to this figure only once, very briefly, without (quite surprisingly) explaining its contents. One wonders why the figure is included in the main manuscript if not much text is devoted to describing it.

Response: Yes, the reviewer is right. We have moved Fig. 3 to Supplementary Fig. 3 as

Supplementary Fig. 3. Combinatorial design details of level-1 structures with different structural motifs. **a** and **b**, Four different types of basic structural motifs for hierarchical uses: the 2R chain-like (a), and three 4R-, 6R-, and 8R-looped bar-link mechanism like structural motif(s) (b). (c) Schematics of the combinatorial design principle of the hierarchical origami-based metastructures: i, Four different types of rigid bar-linkage kinematic mechanisms as structural motifs: the 2R non-loop rigid kinematic mechanism (two rigid links hinged by one rotatable bar), and the 4R, 6R and 8R looped overconstrained rigid kinematic mechanisms (the quantity of rotating bars is equal to the rigid links; ii-iv Schematics of the construction of higher level n ($n \geq 2$) origami-based metastructures by using the same four structural motifs in level 1 structures and combinatorially replacing the higher level links with lower level structures, see the links highlighted by the rectangular shaped dash lines.

Meanwhile, we also update all figures order in manuscript correspondingly.

Figure 4 includes several small insets with small labels /legends that are difficult to read and interpret.

Response: Thanks. All insets in Figure 4 is updated with larger labels and legends as

Fig. 4. Continuous shape morphing of an optimal level-2 structure. a Shape-morphing configurations diagram in the 3D printed prototype exhibiting hierarchical transition tree-like features. Scale bar: 3cm.”

Supplementary notes.

The whole text of the supplementary notes, which seems to be given to demonstrate and explain the differences and novelties of the work, is not well-structured. It seems as though the authors assign to these notes an important role, considering especially their length. However, the paragraphs are not cohesively merged and written in a structured sequence. Several parts are repeated differently, making the text lengthy and unclear. It is not straightforward to appreciate a structure of contents. In addition, there are important paragraphs, such as the one below, where a comparison is made with existing strategies, but no reference is given to them. The lack of references therein makes the comparison unfeasible. The reader is left with no means to verify the authors' claims.

Continuous structural form-based reconfigurable systems. Essentially, these design strategies and our work are two different schemes for reconfigurable and shape morphing structure designs. These methods generally aim for construct functional structures.

Response: We thank the reviewer for the suggestive comment.

We have rewritten the supplementary notes to make it more concise and clear. The repeated sentences and parts are deleted.

Moreover, references are added in the sentence the reviewer mentioned. The remaining parts in supplementary notes are updated with necessary literatures.

SI and manuscript.

Some sentences in the supplementary notes are repeated verbatim in the introduction. I recommend avoiding this practice. The SI should complement and not repeat the text in the main manuscript.

Response: Following the reviewer's suggestion, we rewrite the supplementary information and delete the repeated parts supplementary notes.

In general, the text includes convolute explanations - uneasy to follow, and sometimes compound with English flaws, such as here

- We attribute this to the fact that the hinges have to be uniquely placed with special relative spatial positions to guarantee the level-1 structures with larger number of initial structural DOFs which will result with some structural constraints and hinder the design possibilities..
- flipping the 8-cube level-1 structures is special cases of manipulating the hinge connections at the base level.
- the first category (Category 1) structure is based on the truth that the four level-1 structures (schematically highlighted with blue color in Supplementary Fig. 5B) are positioned with same spatial postures and with both the xz-plane and yz-plane symmetries
- Different with the Category 1
- other level-2 structures can thus be classified with more categories.
- To more clearly see the shape morphing process, we refer to the graph theory applied in computer science where to build ordered data/dataset tree, and construct the corresponding data-tree like diagram applicable for all the proposed hierarchical origami metastructures in this work.
- we find that by pre-assigning a local Cartesian coordinate systems

- a comprehensive view of the continuously evolving shape-morphing configurations diagram > recursive sentence
- Most importantly, once designed, almost all previous works started from this method lack design programmability and deficient for inverse designs? A verb is missing in the second half of the sentence
- these structures exhibit with only one reconfiguration degree of freedom, exhibit with??
- These combinatorial hierarchical mechanisms is
- Comparison between selected four combinatorial categories of level-2 .. on the number

Response: We thank the reviewer for pointing these out and helping us improve our work.

We have corrected and updated the related sentences. Moreover, we have also thoroughly checked the whole text to avoid any confusing expressions and redundant sentences.

Other typos

our work outperform the > outperforms

single reconfiguration degree of freedom, which constrict > constricts

looped kinematic mechanisms has been demonstrated > have

programed >

hybrid architectures by assembly the kinematic cube >> by assembling ?? not sure the meaning here.

advantages: Firstly,

The transition kinematics describe > describes

combinatorically hinging two adjacent cubes at four pair > pairs? But what the term “combinatorically” denotes here

The use of truth and truthfully should be carefully pondered in scientific writing. These terms can be adopted when mathematical proof is given. It doesn't seem that this is the case.

Response: Thanks. All typos and formats mentioned by the reviewer are corrected. We also have thoroughly the main text and Supplementary Information.

Moreover, we modify the original expression of “combinatorically hinging two adjacent cubes at four pair...” into “combinatorically hinging two adjacent cubes at one of the four cube edge pairs” for clarification.

“Combinatorically” here means that between any two adjacent cubes, there are four different choices, i.e., four edges, to place one hinge to connect them together. The details can be checked in Supplementary Fig. 4a-4b, see the **Figure R4** below.

Figure R4 Schematic details of “combinatorically hinging two adjacent cubes” from Supplementary Fig. 4a and 4b.

REVIEWERS' COMMENTS

Reviewer #1 (Remarks to the Author):

The authors have successfully addressed this reviewer's comments.

Reviewer #2 (Remarks to the Author):

The authors have diligently addressed my comments, demonstrating their commitment to improving the paper. They have added important additions and definitions to the text as recommended, streamlined several paragraphs and sentences in the main manuscripts, and completely rewrote entire sections in the SI, showing their responsiveness to constructive feedback. This is appreciated. The paper can be published.